# Visual Agents as Fast and Slow Thinkers

**Guangyan Sun**[♣♠]*, **Mingyu Jin**[♡]*, **Zhenting Wang**[♡], **Cheng-Long Wang**[♦], **Siqi Ma**[°],
**Qifan Wang**[‡], **Tong Geng**[♣], **Ying Nian Wu**[⋈], **Yongfeng Zhang**[♡]†, **Dongfang Liu**[♠]†
[♠] Rochester Institute of Technology [♣] University of Rochester [♡] Rutgers University
[⋈] University of California, Los Angeles [‡] Meta AI [♦] KAUST [°] Westlake University

## Abstract

Achieving human-level intelligence requires refining cognitive distinctions between *System 1* and *System 2* thinking. While contemporary AI, driven by large language models, demonstrates human-like traits, it falls short of genuine cognition. Transitioning from structured benchmarks to real-world scenarios presents challenges for visual agents, often leading to inaccurate and overly confident responses. To address the challenge, we introduce **FAST**, which incorporates the **Fa**st and **S**low **T**hinking mechanism into visual agents. FAST employs a switch adapter to dynamically select between *System 1/2* modes, tailoring the problem-solving approach to different task complexity. It tackles uncertain and unseen objects by adjusting model confidence and integrating new contextual data. With this novel design, we advocate a *flexible system*, *hierarchical reasoning* capabilities, and a *transparent decision-making* pipeline, all of which contribute to its ability to emulate human-like cognitive processes in visual intelligence. Empirical results demonstrate that FAST outperforms various well-known baselines, achieving 80.8% accuracy over $VQA^{v2}$ for visual question answering and 48.7% $GIoU$ score over ReasonSeg for reasoning segmentation, demonstrate FAST's superior performance. Extensive testing validates the efficacy and robustness of FAST's core components, showcasing its potential to advance the development of cognitive visual agents in AI systems. The code is available at this link.

## 1 Introduction

In the field of artificial intelligence, *System 2* delineates a cognitive mode distinguished by deliberate, analytical, and consciously reasoned processes (Wei et al., 2022; Wang et al., 2023d; Zelikman et al., 2022; Zhou et al., 2023; Hua & Zhang, 2022). This mode is juxtaposed to *System 1*, which embodies intuitive, automatic, and unconscious cognition. Achieving human-level intelligence in AI systems necessitates the deliberate cultivation and refinement of these cognitive distinctions. This process is crucial for the development of advanced reasoning and decision-making capabilities (Zhang et al., 2023d;e; Hao et al., 2023).

The emergence of foundation models marks a significant turning point, where Large Language Models (LLMs) based agents have made remarkable strides in many areas, showcasing human-like intelligence across diverse tasks (Brown et al., 2020; Kojima et al., 2022; Ge et al., 2023). However, this achievement is primarily attributed to some features of foundation models: overparameterization

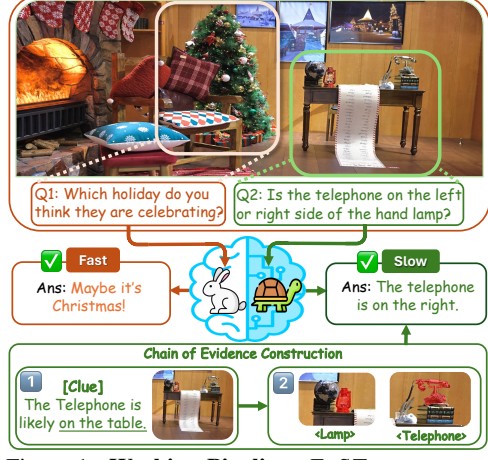

Figure 1. **Working Pipeline.** FAST represents a solution rooted in system switching, demonstrating pronounced capabilities in *hierarchical reasoning* and *ad-hoc explainability*.

and the availability of vast, general-purpose

---

* Equal contribution and shared co-first authorship.
† Corresponding author.

datasets (Kaplan et al., 2020; OpenAI, 2024). It is imperative to note that while these models exhibit human-like traits (*e.g.*, inductive and deductive reasoning (Huang & Chang, 2023b; Dasgupta et al., 2022; Jin et al., 2024b)), these characteristics do not equate to the processes of *System 1/2* thinking (Nye et al., 2021; Yao et al., 2023b) and are far less intelligent than human thinking.

In practice, visual agents often encounter challenges when moving from controlled, structured benchmarks to complex, real-world environments (Wu & Xie, 2024; Ge et al., 2023). This problematic circumstance will result in spurious reasoning pathways, akin to hallucinations, where they struggle to acknowledge their limitations or uncertainties (Gunjal et al., 2024; Chen et al., 2023c). Such an issue arises from the absence of explicit modeling of the fast and slow cognitive processes, reminiscent of human *System 1* and *System 2* thinking (Yao et al., 2023b; Kahneman, 2011). Consequently, when faced with intricate inquiries, the Multimodal Large Language Model (MLLM) frequently offers overly confident yet inaccurate responses (Wu & Xie, 2024; Chen et al., 2024b; Tong et al., 2024a). Addressing this problem entails reassessing MLLM algorithms to incorporate insights from the interplay between fast and slow thinking (*System 1* and *System2*) observed in human cognition. Our design philosophy guides us to incorporate human qualities into our work.

In this study, we introduce the **Fa**st and **S**low **T**hinking (**FAST**) mechanism into visual agents. More concretely, we design a switch adapter to determine whether the encountered problems are best addressed using which thinking mode. Simple tasks require only fast thinking (*System 1*) for a straightforward problem-solving pipeline, while complex tasks necessitate the slow, deliberate processing of *System 2* (see Fig. 1). Specifically, System 2 is triggered when we encounter visual challenges that have: ① *Uncertainty:* When the model has low confidence in directly identifying the object to which the complex query is referring. For example, the query asks "the appliance for storing and cooling food" instead of "refrigerator," and ② *Invisibility:* When dealing with minuscule-sized objects that evade detection by standard visual encoders, where normal visual agents cannot tell what it is. This switch adapter is achieved by designing negative contextual data to re-adjust the model's confidence and ignite world knowledge (as detailed analysis in §3.2). Subsequently, a proposal adapter is engaged to outline regions that are related to the questions. This allows visual agents to leverage the newly acquired data, thereby facilitating a more detailed and precise response. Further, if the inquiry necessitates detailed insights into particular instances, a seg adapter provides segmentation masks, offering additional contextual information for deeper analysis (as detailed analysis in §3.3).

**FAST** enjoys a few attractive qualities. ❶ **Flexible system**: Building on a foundation that explicitly models *System 1/2 thinking*, our proposed method adeptly handles complex visual tasks, demonstrating competitive performance in a streamlined pipeline (see §2.1). FAST's core epistemology combines an intuitive mechanism for straightforward cases with deliberate analytics for more intricate scenarios, thereby enhancing the development of a human-like visual agent. ❷ **Hierarchical reasoning**: FAST perceives visual tasks with a top-down granularity, encompassing image-level cues, box-level candidates, and pixel-level targets (see Fig. 2). This progressive approach facilitates a sensible understanding of visual content, starting from global concepts, progressing through region-specific candidate assessment, and culminating in precise target identification. Each stage involves developing concrete ideas and establishing a coherent "*chain of evidence*" to support the final inference. ❸ **Transparent pipeline**: FAST's decision-making process embodies a neuro-symbolic essence in System 2 mode, yielding intermediate step outputs as interpretable symbols (*e.g.*, bounding boxes or masks), facilitating direct visual inspection by humans. This inherent reasoning mechanism enables *ad-hoc explainability* of the model's behavior (see Fig. 3), distinguishing FAST from prior approaches (Liu et al., 2024a) that lack precise explication of their operational mechanisms.

We conducted a series of experiments to validate the efficacy of our proposed method. In §3.1.1, we apply FAST to visual question answering and multimodal benchmarks. FAST demonstrates significantly improved performance over baselines such as LLaVA-v1.5 (Liu et al., 2024a), achieving performance gains on benchmarks like TextVQA (Singh et al., 2019) with a 2.5% increase in accuracy and a total score improvement of 6.7 on MME (Fu et al., 2024). In §3.1.2, we explore the versatility of our approach through its application to tasks such as referring and reasoning segmentation, with performance gains including an increase of 4.1% $CIoU$ with LLaVA-v1.5, and improvements of 3.2% $CIoU$ and 2.7% $GIoU$ on the ReasonSeg dataset over LISA-7B (Lai et al., 2024). The robustness and effectiveness of the core components of our FAST framework are further substantiated through a series of ablation studies, as elaborated in §3.3.

## 2 METHODS

**Notation.** The integration of components in visual agents $\mathcal{F}$ (based on the Large Language Model) typically involves a visual encoder, denoted as $\mathcal{E}_V$, a nature language encoder, represented by $\mathcal{E}_L$, and a Language Language Model such as Vicuna (Chiang et al., 2023). Initially, the visual agent is presented with an image $\mathcal{I}$ and an accompanying textual prompt $\mathcal{Q}$, which could be a question or instruction. Then the visual agent combines these multimodal tokens into a united space. Finally, the visual agent outputs a textual response $R$ given the textual and image input. The generation process can be expressed as Equation 1:

$$R = \mathcal{F}\left[\mathcal{E}_V(I), \mathcal{E}_L(Q)\right] \tag{1}$$

**Definition 1** *(System 1 and System 2) System 1 and 2 are two different systems of thinking proposed by Nobel Laureate Daniel Kahneman in his book Thinking, Fast and Slow (Kahneman, 2011).*

*System 1 (Fast Thinking): Unconscious, automated thinking processes, fast, intuitive, effortless responsible for automatic responses and basic cognitive operations in daily activities, vulnerable to heuristic biases and errors, e.g., recognizing familiar faces, and knowing the location of objects.*

*System 2 (Slow Thinking): Conscious, energetic thinking processes, slow, effortful, logical, and analytical, responsible for complex calculations, reasoning, and decision-making, can monitor and control System 1 processes, e.g. filling out a tax form, finding the position of a word in a sentence.*

### 2.1 FAST

We present FAST (see Fig. 2), a novel framework designed to efficiently handle both simple and complex visual queries. FAST features a dynamic system switch mechanism that enables rapid responses to straightforward questions (*System 1*) and accommodates deliberate reasoning for intricate scenarios (*System 2*). During slow thinking, the system uses contextual clues to identify a relevant region, facilitated by a proposal adapter. The adapter generates a bounding box around the target object, and if needed, a pixel-level mask adapter refines the proposal for further details. Finally, we summarize the gathered information from the whole system to provide a comprehensive answer.

**System Switch.** Current works on visual agents mostly rely on visual question-answering data, which gives direct answers (*System 1*) after inquiry as Eq. 1. However, attempting to answer questions directly in this way can compromise the reliability of the responses. Agents tend to hallucinate over questions that require more deliberate reasoning and visual details. To reduce hallucination and make the model reliable, we utilize a system switch trigger to tell when to require more visual information. Specifically, for a question $\mathcal{Q}$ and an image $\mathcal{I}$, we define a MLLM with switch adapter $\mathcal{S}$ and formulate the fast $\mathcal{F}_{fast}$ and slow thinking process $\mathcal{F}_{slow}$. When the query is easy, the frame does not need the switch adapter $\mathcal{S}_{adapter}$ and only output result $\mathcal{R}$ by $\mathcal{F}_{fast}$ as Equation 2.

$$\mathcal{R} = \mathcal{F}_{fast}\left[\mathcal{E}_V(I), \mathcal{E}_L(Q)\right] \tag{2}$$

**Remark 2.1** *(Switching-friendly dataset) A Negative Data for Target Objects Reasoning Dataset $\mathcal{D}$ of 100,000 (image, question, answer) triples was constructed to facilitate the identification of target regions or objects required to answer a question. The dataset constructs questions about the absence or details of certain objects, deliberately made too small to be perceived by the visual encoder. Section A for more details.*

**Remark 2.2** *(Switch Adapter) A light-weight adapter that is fine-tuned with both positive fast-thinking data and negative data (Remark. 2.1) to acquire system switching capability. When the adapter encounters harder questions, the switch mechanism will be triggered for later slow thinking.*

Note that a slow thinking process is not always activated. The system switch adapter as Remark. 2.2 $\mathcal{S}_{adapter}$ will determine whether the question for the particular image is sufficient to give a direct answer. If so, the fast mode $\mathcal{F}_{fast}$ will give a quick and direct response as Equation 2. If there is any missing information about the question that current agent cannot solve, the switch adapter will be activated and find the pattern to elicit all the possible missing objects $\mathcal{O}_{missing}$ related to question and context clues $\mathcal{C}_{clue}$ which is the possible location of the missing objects as Equation 3.

$$\mathcal{O}_{missing}, \mathcal{C}_{clue} = \mathcal{S}_{adapter}\left[\mathcal{E}_V(I), \mathcal{E}_L(Q)\right] \tag{3}$$

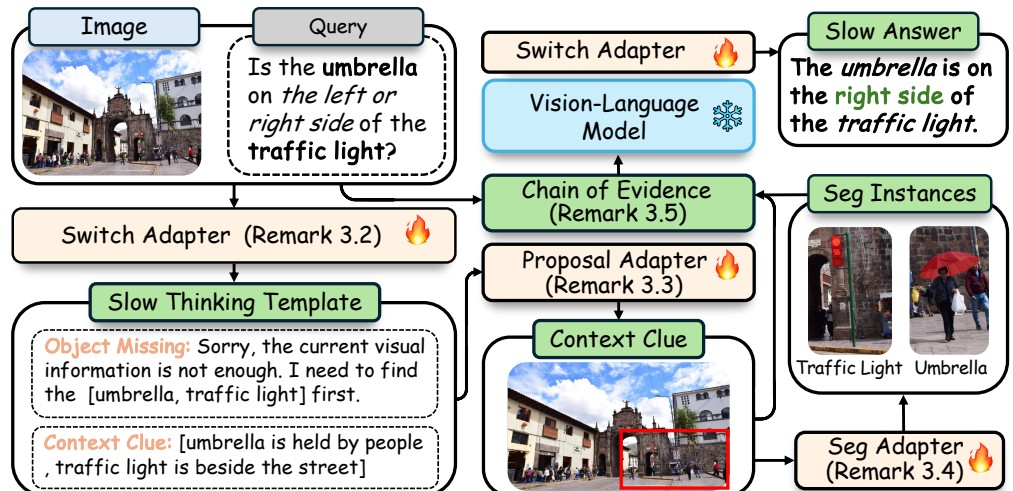

Figure 2. **Slow Thinking Mode of FAST.** Our *slow thinking mode* comprises three core modules: *Switch Adapter*, which selectively activates a slow and analytical thinking mode when encountering complex visual queries, supplementing with extensive world knowledge to provide missing objects and contextual clues; *Proposal Adapter*, which identifies and emphasizes regions of interest within the visual inputs; *Seg Adapter*, which delivers precise pixel-level segmentation, enhancing depth of the visual analysis. The outputs from each module are integrated into a *chain of evidence* (see Fig. 3), providing a methodical and accurate response. FAST represents a neural-symbolic approach that combines the strengths of symbolic reasoning, ensuring that our system is effective and interpretable.

Specifically, we use negative data that contain missing objects $\mathcal{O}_{missing}$ and context clues $\mathcal{C}_{clue}$ for training the system switch adapter for triggering the slow mode as Remark. 2.2. The slow mode should deal with question-image pairs that are 1) uncertain in pinpointing the specific object in question, and 2) too small to perceive for the standard visual encoder. So we utilize triplet data in dataset Remark. 2.1 (image, question, answer) as where the question requires objects that are not in the image or too small to be perceived by the visual encoder. The threshold is set to be $20 \times 20$. We require the model to tell that certain objects are missing instead of a direct answer, and we utilize the world knowledge to list all the objects and also the context information for later deliberate reasoning.

## 2.2 PRELIMINARY

**Hierarchical Reasoning.** We use a top-down scheme to reason over multi-scale granularity images effectively in order to reason and take advantage of world knowledge progressively. Similar to humans would look for some context clue to find specific objects relating to questions and zoom in if they think the answer lies in a particular region, we model this process with system switch adapters as Remark. 2.2 to focus on the context clue $\mathcal{C}_{clue}$ generated from the switch adapter as Equation 3.

We denote the MLLM as a proposal adapter $\mathcal{P}_{adapter}$ (visual agent). In *System 2*, FaST uses many visual agents to accomplish hierarchical reasoning. The frame tries to narrow down the search space by using the question $\mathcal{Q}$ and the previously obtained clue $\mathcal{C}_{clue}$ to let the proposal adapter output a region $Region$ that aligns with the question and the context clue as Equation 4.

$$Region = \mathcal{P}_{adapter}\left[\mathcal{E}_V(I), \mathcal{E}_L(Q), \mathcal{C}_{clue}\right] \tag{4}$$

After getting the region, the visual agents $\mathcal{P}_{adapter}$ will be asked to focus on a more specific target with a bounding box $[Bboxes]$ complemented by the context clue $\mathcal{C}_{clue}$ and region $Region$ get from Equation 4. This process can reveal the step-by-step reasoning and be modeled as Equation 5.

$$[Bboxes] = \mathcal{P}_{adapter}\left[\mathcal{E}_L(Q), Region, \mathcal{C}_{clue}\right] \tag{5}$$

**Remark 2.3** *(Proposal Adapter) A lightweight adapter that is fine-tuned with proposal data to acquire the capability of finding the corresponding region given the context clue or object name.*

**Remark 2.4** *(Pixel-level mask decoder) The Pixel-level mask decoder is the decoder of segment anything(SAM (Kirillov et al., 2023)). The pixel-level mask decoder is fine-tuned to produce target masks based on the hidden embeddings.*

When we have a more specific target proposal(bounding box $[Bboxes]$), FAST will apply a fine-grained pixel-level mask decoder $\mathcal{P}_{Seg}$ as Equation 6 to output the specific mask part $[Mask]$ of the target proposal $[Bboxes]$ to focus on as Equation 6. We name this whole process from $Region$ to $[Mask]$ *chain of evidence* as Remark. 2.5 similar to thinking more and more deeply by humans.

$$[Mask] = \mathcal{P}_{seg}\left[\mathcal{E}_L(Q), [Bboxes], \mathcal{O}_{missing}\right] \tag{6}$$

**Remark 2.5** *(Chain of Evidence) Chain of evidence is like the chain of thought in a Large Language Model. But we define it as a deeper and deeper step of thinking based on correct evidence in our frame* FAST. *The completion of the chain of evidence needs many visual agents to work together.*

After getting the target proposal (bounding box $[Bboxes]$) from context clue $\mathcal{C}_{clue}$ with proposal adapter and specific mask part $[Mask]$ by missing objects with seg adapter, a *chain of evidence* is constructed as Remark 2.5 and Fig. 3. Our FAST framework then summarizes all this information ($\mathcal{I}$ and $\mathcal{Q}$) and the *chain of evidence* with switch adapter to give the final correct reasoning answer $Ans$ as Equation 7:

$$Ans = \mathcal{F}_{Slow}\left[\mathcal{E}_L(Q), \mathcal{E}_V(I), [Bboxes/Mask]\right] \tag{7}$$

The decision-making process in FAST is distinguished by its neuro-symbolic nature, which generates intermediate outputs as easily interpretable symbols, including region-of-interest (RoI) driven boxes and object-driven masks. This capability allows humans to perform direct visual inspections, thereby augmenting the transparency of the model's operations. Moreover, the intrinsic reasoning mechanism of FAST enhances the ad-hoc explainability of its behavior, see Fig. 2.

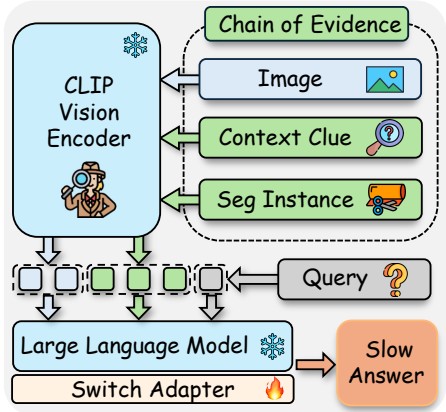

Figure 3. **Chain of Evidence**. FAST represents a solution rooted in switching, demonstrating pronounced capabilities in *hierarchical reasoning* and *ad-hoc explainability*.

## 2.3 IMPLEMENTATION DETAILS

The framework of FAST(as Fig. 2)'s implementation details are shown in this section below.

• *Visual Agents.* We choose the architecture and configuration of LLaVA-v1.5 (Liu et al., 2024a) as our visual agent. The most important component in a visual agent is the visual encoder $\mathcal{E}_V(I)$: A *CLIP-ViT-L-336px* model (Radford et al., 2021) is used, where input images are resized or padded to $336 * 336$ pixels, learning to associate visual features with corresponding textual descriptions. An MLP projection with channels of $[256, 4096, 4096]$ is used for connecting image representations into the word embedding space.

• *Mask Decoder.* The mask decoder $\mathcal{P}_{seg}$ architecture is identical to SAM. Besides, it is fully fine-tuned with a collection of semantic segmentation (Caesar et al., 2018; Zhou et al., 2017; Ramanathan et al., 2023; He et al., 2022; Chen et al., 2014) and referring segmentation (Mao et al., 2016; Kazemzadeh et al., 2014) datasets to efficiently map the <seg>token representations to a mask if the FAST need to segment.

• *Chain of Evidence.* When we apply the *chain of evidence* as Remark 2.5 in the LLM to get the answer as the final step like Equation 7. The whole sequence of the *chain of evidence* is too long to load in the $\mathcal{F}_{Slow}$. So FAST needs a visual sampler based on cross-attention that is trained to decrease the number of image tokens to a suitable length (from 256 to 32), apart from MLP projection.

## 3 EXPERIMENT

We utilize eight popular benchmarks to evaluate our framework **FAST** comprehensively, categorized into general visual question answering (VQA) datasets and multimodal benchmarks. The

VQA benchmarks include VQA-v2 (Goyal et al., 2017), GQA (Hudson & Manning, 2019), ScienceQA (Lu et al., 2022), and TextVQA (Singh et al., 2019) which focus on optical character recognition. For multimodal benchmarks evaluation, we use the hallucination benchmark POPE (Li et al., 2023c), along with comprehensive benchmarks such as MME (Fu et al., 2024), MM-Vet (Yu et al., 2024), and SEED (Li et al., 2024). We compare our model with the baseline LLaVA-v1.5 (Liu et al., 2023a), and other multimodal large language models. To thoroughly assess our model's understanding of pixel-level instances, we evaluate its performance on referring segmentation and grounding benchmarks, including refCOCO (Kazemzadeh et al., 2014), refCOCO+ (Kazemzadeh et al., 2014), and refCOCOg (Caesar et al., 2018). Further, to examine the model's reasoning capabilities on FASST framework, we consider the Reasoning Segmentation benchmark (Lai et al., 2024).

## 3.1 MAIN RESULTS

### 3.1.1 EXPERIMENTS ON VQA AND MULTIMODAL BENCHMARKS

**Training.** In developing the Switch Adapter, we employed the LLaVA-v1.5 (Liu et al., 2024a) framework, conforming strictly to its established training protocols. We incorporated negative samples from $V^*$ (Wu & Xie, 2024) with contextual cues to enhance system switching capability to amplify multimodal inferential and world knowledge. This augmented dataset was combined with LLaVA-v1.5's supervised dataset and trained for one epoch. For the Proposal Adapter, we augmented the LLaVA-v1.5 dataset with region-specific bounding boxes based on contextual cues and queries, then fine-tuned for one epoch to optimize proposal generation. The Segmentation Adapter utilized the LISA (Lai et al., 2024) architecture integrated with the LLaVA-v1.5, employing SAM as the mask decoder. The adapter was fine-tuned using the same datasets as Lisa, including semantic segmentation, referring segmentation, and reasoning segmentation. This fine-tuning process involved 10,000 steps to improve the model's segmentation capabilities. Throughout developing the Switch Adapter, Proposal Adapter, and Segmentation Adapter, we employed the LoRA (Low-Rank Adaptation) technique (Hu et al., 2022). By leveraging LoRA, we introduce minimal additional parameters while preserving the original multimodal large language model's architecture and efficiency. All experiments used 8 NVIDIA TESLA A100-80GB GPUs.

**Metric.** In model evaluation across diverse datasets, various performance metrics are utilized.

*Accuracy.* The primary evaluation metric utilized in the $VQA^{v2}$, GQA, TextVQA, ScienceQA, and SEED benchmarks is accuracy. Accuracy is a performance measure that quantifies the exact match percentage between predicted and acceptable ground truth answers, indicating a model's precision.

*F1 Score.* The POPE dataset uses the F1 Score to balance precision and recall, providing a comprehensive assessment by harmonizing the trade-off between positive prediction accuracy and recall.

*Total Score.* The MME evaluation metrics include accuracy (based on individual questions) and accuracy+ (considering both questions per image), reflecting a stricter and more comprehensive model understanding. Random accuracies for these metrics are 50% and 25%, respectively. Perception scores, calculated as the sum of these metrics across subtasks, total 2000 for perception.

*GPT-Evaluation.* In the MM-Vet dataset, performance is evaluated by GPT-4 through a comparative analysis of predicted and ground truth answers, generating a score to quantify alignment.

**Results.** As depicted in Table 1, FASST demonstrates superior performance across multiple VQA datasets and multimodal benchmarks when compared to established methods. To ensure fairness in comparison, all methods in Table 1 share the same visual encoder: basic CLIP (Radford et al., 2021). Remarkably, FASST consistently surpasses the LLaVA-v1.5 model, achieving significant improvements in performance across all evaluated datasets. Specifically, in VQA datasets, our model outperforms LLaVA-v1.5 by 2.3% in $VQA^{v2}$, 1.8% in GQA, and 2.5% in $VQA^T$. Ad-

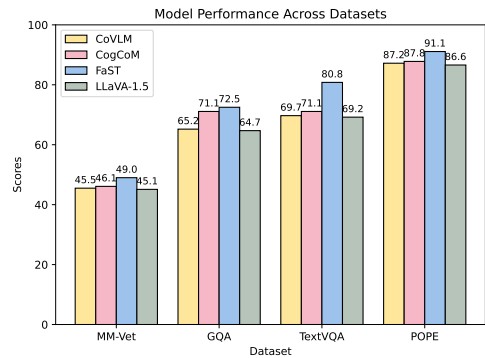

Figure 4. **The Comparison with CoVLM and CogCoM.** These models use the more powerful vison encoder.

| Method | LLM | VQA Datasets | | | | Multimodal Benchmarks | | | |
|---|---|---|---|---|---|---|---|---|---|
| | | $VQA^{v2}$ | GQA | $VQA^T$ | $SQA^I$ | POPE | MME | SEED | MM-Vet |
| BLIP-2[ICML23] | Vicuna-13B | 65.0 | 32.3 | 42.5 | 61.0 | 85.3 | 1293.8 | 46.4 | 22.4 |
| InstructBLIP[NeurIPS24] | Vicuna-13B | - | 49.5 | 50.7 | 63.1 | 78.9 | 1212.8 | 53.4 | 25.6 |
| Qwen-VL-Chat[arXiv23] | Qwen-7B | 78.2 | 57.5 | 61.5 | 68.2 | - | 1487.5 | 58.2 | - |
| mPLUG-Owl2[CVPR24] | LLaMA-7B | 79.4 | 56.1 | 58.2 | 68.7 | - | 1450.2 | **61.6** | **36.2** |
| Monkey[CVPR24] | Qwen-7B | 80.3 | 60.7 | - | **69.4** | 67.6 | - | - | - |
| LLaVA-v1.5[CVPR24] | Vicuna-7B | 78.5 | 62.0 | 58.2 | 66.8 | 85.9 | 1510.7 | 58.6 | 30.5 |
| Chain of Spot [arXiv24] | Vicuna-7B | 80.7 | 63.7 | 60.9 | 68.2 | 86.4 | 1501.1 | 59.7 | 30.8 |
| V*[CVPR24] | Vicuna-7B | - | - | - | - | 82.4 | 1128.9 | 41.7 | 27.7 |
| Visual CoT[arXiv24] | Vicuna-7B | - | 63.1 | **77.5** | - | - | - | - | - |
| FAST (Ours) | Vicuna-7B | **80.8** | **63.8** | 60.7 | 68.9 | **86.4** | **1517.4** | 60.1 | 31.0 |
| Δ (vs LLaVA-v1.5) | Vicuna-7B | +2.3 | +1.8 | + 2.5 | +2.1 | +0.4 | +6.7 | + 1.5 | + 0.5 |

Table 1. **Main results on eight VQA and multimodal benchmarks.** Our FAST consistently outperforms the baseline LLaVA1.5 model across all evaluated benchmarks, denoted with line Δ.

ditionally, FAST excels in multimodal benchmarks, with notable increases of 6.7 in the MME score, 1.5 in the SEED score, and 0.5 in the MM-Vet score, highlighting its versatility and effectiveness in handling a broad range of domains. These results underscore the robustness of FAST, particularly in tackling complex visual and textual tasks. Moreover, Fig. 4 showcases a direct comparison between FAST, CoVLM (Wang et al., 2023b), and CogCoM (Qi et al., 2024), both of which employ the more powerful EVA2-CLIP-E (Sun et al., 2023) model as their visual encoder. As expected, these models exhibit stronger performance due to their enhanced encoder. To align with this, we replaced our original vision encoder with EVA2-CLIP-E, which resulted in further improved performance, ensuring a more rigorous and fair comparison with state-of-the-art methods.

### 3.1.2 EXPERIMENTS ON REFERRING AND REASONING SEGMENTATION

**Training.** The training settings for the Switch Adapter and Proposal Adapter remain consistent with those previously described as §3.1.1. During the training phase of the Segmentation Adapter, certain specific datasets are intentionally omitted to uphold an unbiased evaluation of referring and reasoning segmentation datasets. This strategic exclusion is a crucial measure implemented to prevent any potential data leakage, thereby ensuring the integrity and reliability of the evaluation results.

**Metric.** Following prior research on segmentation (Kazemzadeh et al., 2014; Mao et al., 2016), two evaluation metrics are employed: Generalized Intersection over Union ($GIoU$) and complete Intersection over Union ($CIoU$).

$CIoU$. The $CIoU$ is calculated based on the cumulative intersection over the cumulative union across all images in the dataset. This approach can introduce a significant bias towards larger objects or images with more objects, as they contribute more to the cumulative union area.

$GIoU$. The $GIoU$ is computed as the average per image $IoU$, where the $IoU$ is calculated for each image, and then the average is taken across all images in the dataset. This metric provides a balanced assessment by treating all images equally, regardless of their size or the number of objects.

**Results.** Table 2 illustrates the performance of FAST compared to recent visual agents like LISA on referring and reasoning segmentation benchmarks. FAST notably outperforms LISA-7B on the refCOCO+ and refCOCOg benchmarks by 2.0% and 0.6% $CIoU$, respectively. For the more complex reasoning segmentation task, FAST shows even stronger results, with a 3.2% $CIoU$ gain and a

| Method | Referring Segmentation | | | Reasoning Segmentation | |
|---|---|---|---|---|---|
| | refCOCO | refCOCO+ | refCOCOg | ReasoSeg | |
| | CIoU | CIoU | CIoU | CIoU | GIoU |
| LAVT[CVPR22] | 72.7 | 62.1 | 61.2 | - | - |
| OVSeg[CVPR23] | - | - | - | 28.5 | 18.6 |
| GRES[CVPR23] | 73.8 | **66.0** | 65.0 | 22.4 | 19.9 |
| X-Decoder[CVPR23] | - | - | 64.6 | 22.6 | 17.9 |
| SEEM[NeurIPS24] | - | - | 65.7 | 25.5 | 21.2 |
| LISA-7B[CVPR24] | **74.1** | 62.4 | 66.4 | 44.4 | 46.0 |
| LLaVA w Seg Adapter | 70.8 | 57.5 | 64.0 | 43.0 | 41.0 |
| FAST (Ours) | 73.3 | 64.4 | **67.0** | **47.6** | **48.7** |

Table 2. **Main results on referring and reasoning segmentation benchmarks.** Our FAST exhibits competitive results in referring segmentation tasks like refCOCOg+ while showcasing superior performance in reasoning segmentation, particularly when evaluated against LISA-7B.

2.7% $GIoU$ improvement over LISA. The results highlight FAST's superior performance and its robustness in handling both straightforward and complex visual reasoning segmentation benchmarks.

## 3.2 ANALYSIS OF SYSTEM SWITCHING ADAPTER

Our study investigates the efficacy of the switch adapter mechanism in balancing accuracy and computational efficiency. As depicted in Fig.5, our analysis illustrates the system's adeptness in discerning between the *System 1* and *System 2* cognitive modes triggered by query complexity. For queries requiring System 2 reasoning, the adapter dynamically combines *System 1* reasoning for simpler subcomponents with *System 2* reasoning for more complex aspects. Consequently, the reported accuracy rates under *System 2* mode (52.2% for MME and 56.8% for GQA) reflect a combination

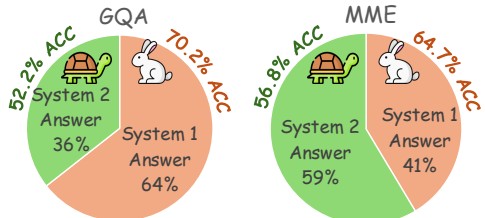

Figure 5. *System 1* **Mode Analysis.** We investigate the system switching ratio, along with fast thinking performance on easy or hard queries defined by the switch adapter.

of reasoning outcomes, emphasizing the adapter's ability to differentiate query complexities and optimize task performance accordingly. This highlights the importance of maintaining *System 1* reasoning for prompt and confident responses while effectively utilizing *System 2* reasoning for complex problem-solving.

Table 3 compares runtime across system configurations. *System 1 Only*, using a switch adapter, operates efficiently with one-time inference, while *System 2 Only*, which constructs a *chain of evidence* for every query, is significantly more resource-intensive. In contrast, FAST balances efficiency and performance, running 31% faster than *System 2 Only*

| Method | MME | | GQA | |
|---|---|---|---|---|
| | Runtime | Result | Runtime | Result |
| *System 1 Only* | 734ms | 1508.7 | 737ms | 61.9 |
| *System 2 Only* | 2938ms | 1518.6 | 2937ms | 64.0 |
| **OURS** | 2023ms | 1517.4 | 1475ms | 63.8 |

Table 3. **Runtime Analysis and Comparison** on only *System 1* (fast), our FAST and only *System 2* (slow).

on MME and 50% faster on GQA, with comparable results. This highlights FAST's ability to optimize cognitive task processing while conserving computational resources.

## 3.3 ABLATION STUDY

| Algorithm Component | GQA | POPE | MME |
|---|---|---|---|
| BASELINE | 62.1 | 85.7 | 1509.2 |
| + Proposal Adapter | 63.2 | 86.0 | 1516.5 |
| + Seg Adpater | 62.8 | 85.8 | 1514.4 |
| **OURS** (**both**) | 63.8 | 86.2 | 1517.4 |

Table 4. **Key Component** Analysis

| Output Component | MME | refCOCOg |
|---|---|---|
| BASELINE* | 1511.8 | 66.0 |
| + Missing Objects | 1513.4 | 66.8 |
| + Context Clue | 1516.6 | 66.4 |
| **OURS** (**both**) | 1517.4 | 67.0 |

Table 5. **Switch Adapter** Output Analysis

**Key Component Analysis.** We undertake a detailed investigation into the core elements of our novel framework, FAST, with particular emphasis on the proposal adapter for contextual region localization and the seg adapter for pixel-level mask segmentation. To establish a comparative baseline, we design a model configuration that excludes both the proposal and seg adapters, instead relying solely on a switch adapter to provide missing objects and context clues. This baseline model serves as the foundation for evaluating the impact of the individual and combined components of the framework. As demonstrated in Table 4, the introduction of the proposal adapter, the seg adapter, or both, results in progressive and substantial improvements in performance across various evaluation metrics. For instance, accuracy on the $VQA^{v2}$ dataset improves from 62.1% to 63.8%, showcasing the considerable value these components add. This underscores the pivotal roles of the proposal and seg adapters in enhancing the model's overall capability, further affirming their importance within the FAST framework.

Further, we evaluate the switch adapter's role in incorporating missing objects and context clues using a variant BASELINE*, which omits these features. Table 5 shows that adding missing objects or context clues improves metrics like MME and refCOCOg, with the best performance achieved

when both are included. These results confirm the importance of all components in optimizing FAST's effectiveness.

## 3.4 QUALITATIVE COMPARISONS OF FAST

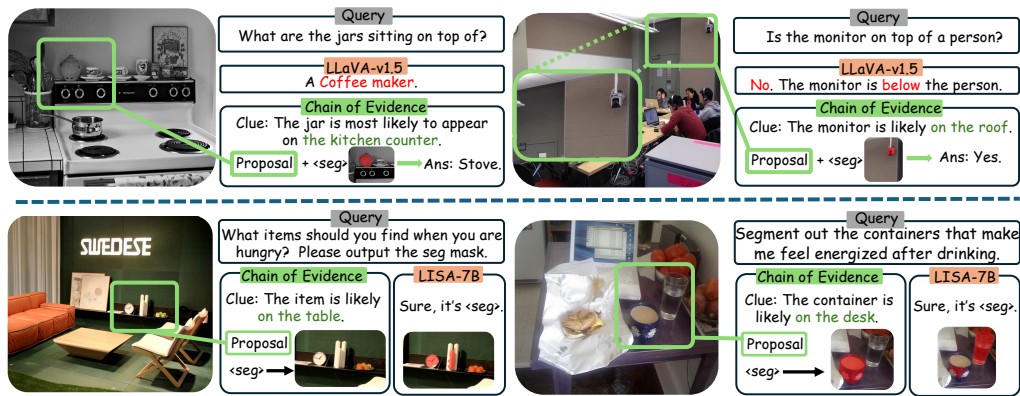

Figure 6. **Qualitative Comparisons of FAST.** The top row shows the VQA results on FAST compared to LLaVA-v1.5. The bottom row presents the segmentation results compared to LISA-7B.

In Fig.6, we present qualitative comparisons that highlight the enhancements introduced by FAST. The top row, above the dotted line, shows results from the VQA task, comparing FAST with LLaVA-v1.5. LLaVA-v1.5 often fails to focus on key areas within the image, leading to incorrect or incomplete responses. In contrast, FAST builds a *chain of evidence* by identifying key objects and elements (e.g., detecting a woman on the street or a monitor on the roof) and then applying object-level pixel masks to accurately determine the focus areas. This enables FAST to provide more precise and deliberate answers. The bottom row, below the dotted line, shows segmentation results, comparing FAST with LISA-7B. LISA-7B struggles with segmenting smaller objects or those requiring more complex reasoning, often causing confusion. In contrast, FAST excels at isolating relevant objects with greater accuracy and granularity, particularly with smaller or less obvious items. This demonstrates FAST's superior performance in both VQA and segmentation tasks, showcasing its ability to handle a wide range of visual and reasoning challenges more effectively than its counterparts.

## 4 RELATED WORKS

**LLM as Visual Agents.** With the capabilities that LLMs have demonstrated in language understanding and generation (Ouyang et al., 2022; OpenAI, 2024; Zheng et al., 2023; Touvron et al., 2023a;b; Wang et al., 2024a; Hua et al., 2024; Mei et al., 2024; Jin et al., 2024a; Chen et al., 2024c), the research community has progressed to explore how LLMs can be enhanced with vision input for multimodal tasks as visual agents (Alayrac et al., 2022; Driess et al., 2023; Li et al., 2023a; Ge et al., 2023; Dai et al., 2024; Liu et al., 2023b; 2024a; Lin et al., 2024; Wang et al., 2024b). There are two paradigms for LLM-based visual agents: end-to-end based and tool-using visual agents. Following the principle of instruction tuning, end-to-end visual agents are trained with a curated visual instruction tuning dataset to digest features from multi-modality, unlocking the capability to answer visual questions (Huang et al., 2023; Luo et al., 2023; Zhu et al., 2023a; Bai et al., 2023; Zhang et al., 2023b;c; Chen et al., 2024a; Ye et al., 2023; Singh et al., 2019). For other visual tasks (*e.g.*, Segmentation, Detection, etc), end-to-end trained tailored agents can further perform downstream tasks (Pi et al., 2023; Peng et al., 2024; Lai et al., 2024; Chen et al., 2023b; Wang et al., 2023c; Dai et al., 2024; Wang et al., 2024c; 2023b; Jiang et al., 2023; Chen et al., 2023a; Zeng et al., 2024). Recent research has focused on leveraging improved vision encoders and fostering more detailed visual understanding, yielding promising results (Fan et al., 2024; Xu et al., 2024a; Shi et al., 2024). While these approaches can be implemented with direct instruction tuning data, they represent a '**System 1**' type of training. This type of training primarily relies on the dataset's quality and tends to provide direct answers that are prone to hallucinations, a consequence inherent to the nature of *System 1* instruction tuning data. For the second paradigm, tool-using models are built on top of a frozen LLM with access to pretrained visual perception tools (Surís et al., 2023; Shen et al., 2023; Lu et al., 2023a). In this scenario, the LLM first selects visual tools and then decides by thoroughly

analyzing the fine-grained information extracted by visual tools (Lu et al., 2023a; You et al., 2023; Wu et al., 2024). While external visual tools enhance the interpretability of the reasoning process, their complexity can introduce inaccuracies. Moreover, the abundance of information generated during reasoning may overshadow key details relevant to the query, resulting in incorrect answers.

Our research introduces a novel and adaptable framework designed to enhance response accuracy by adopting distinct slow thinking cognitive modes. Unlike traditional end-to-end visual agents, our framework, FAST, systematically assesses information sufficiency, thereby mitigating the risk of overconfidence. When *System 2* (slow, analytical thinking) is activated, FAST employs multiple experts to construct a coherent *chain of evidence*. This approach ensures the generation of accurate and interpretable responses, significantly advancing the reliability and transparency of visual agents.

**System 2 in AI.** Recently, LLMs have been engineered to produce text that mimics the step-by-step reasoning process characteristic of human cognition, akin to the analytical and deliberate thought processes associated with what is termed as *System 2* in the human cognition process (Qiao et al., 2023; Huang & Chang, 2023a; Wang et al., 2023a; Shaikh et al., 2023; Shao et al., 2024). The systematic approach to problem-solving is a hallmark across various domains, including mathematical word problems (Kojima et al., 2022; Wang et al., 2023d; Lightman et al., 2023; Cobbe et al., 2021; Liu et al., 2023c; Zhu et al., 2023b; Lu et al., 2023b), logical reasoning (Yao et al., 2023d;a; Besta et al., 2024; Wen et al., 2023; Lei et al., 2023; Cheng et al., 2024; Jin et al., 2024b), and multi-modal reasoning (Chen et al., 2024e; You et al., 2023; Wu & Xie, 2024). In Explainable AI, this systematic method is emulated by the model as it generates a text-based elucidation of its reasoning and decision-making process through step by step reasoning process (*e.g.*, chain of thought) (Han et al., 2024; Zhao et al., 2024; Jacovi & Goldberg, 2020; Hua & Zhang, 2022). However, it is crucial to recognize that LLMs, while powerful, are not exempt from encountering challenges when facing complex problems. One such challenge is the issue of hallucination (Zhou et al., 2024; Cui et al., 2023; Li et al., 2023b; Zhang et al., 2023a; Chen et al., 2024d; Guan et al., 2024a), which can distort the model's reasoning process and lead to inaccuracies in the explanations provided. Initially, LLM reasoning is seen as only a linear chain of thoughts, where each step in the reasoning process is clearly articulated. As models evolve, they adopt more complex structures like hierarchical trees (Geng et al., 2023; Yao et al., 2023b) and intricate graphs (Besta et al., 2024), which enable them to handle much more complex problems but also restrict their general applicability because of increased topological complexity (Yao et al., 2023b;a; Besta et al., 2024; Lei et al., 2023; Wen et al., 2023). Moreover, these complex structures can lead to errors propagating through the model's reasoning, causing a cascade of mistakes (Xu et al., 2024b). To counter this, incorporating feedback from intermediate reasoning steps and employing iterative refinement, which is similar to human reflection, could help mitigate errors (Chu et al., 2023; Tong et al., 2024b; Guan et al., 2024b; Madaan et al., 2023; Yuan et al., 2024; Wu & Xie, 2024). In unsupervised scenarios, such feedback is vital for enhancing the reasoning capabilities of LLMs and reducing errors (Yao et al., 2023c).

Our key contribution is the introduction of the *chain of evidence* within multimodal reasoning frameworks. This methodology enriches each reasoning step with accurate, image-based cascading information, effectively mirroring human visual and *System 2* cognitive processes. Our approach enhances accuracy and significantly improves interpretability and generalization capabilities.

## 5 CONCLUSION

In this study, we introduced FAST, a framework that combines *System 1* (which is fast and intuitive) and *System 2* (which is slow and deliberate) thinking to improve visual agents' reasoning and decision-making. FAST adapts to queries of varying complexity with a flexible system switch, delivering quick responses for simple tasks and using hierarchical reasoning for more complex scenarios. The FAST leverages neuro-symbolic decision-making transparent pipeline delivering interpretable intermediate outputs that enable explainability. Our results show significant improvements across benchmarks, demonstrating the effectiveness of FAST's *chain of evidence* in reducing hallucinations and improving interpretability. Furthermore, ablation studies highlight the critical importance of contextual clues, symbolic reasoning, and pixel-level adapters in refining visual reasoning and understanding, marking a step forward in creating more reliable and accurate AI cognition.

## 6 ACKNOWLEDGMENTS

This research was supported by the National Science Foundation under Grant No.2242243. The views and conclusions contained herein are those of the authors and should not be interpreted as necessarily representing the official policies or endorsements, either expressed or implied, of U.S. Naval Research Laboratory (NRL) or the U.S. Government. The U.S. Government is authorized to reproduce and distribute reprints for Government purposes notwithstanding any copyright notation herein.

## 7 ETHICAL SAFEGUARDS

In our paper introducing a novel framework FAST, we implement rigorous ethical measures to prevent potential misuse and promote responsible application. These measures are delineated in comprehensive protocols accompanying the final release of models and datasets. Our protocols encompass stringent usage guidelines, access controls, incorporation of safety filters, and monitoring systems. These concerted efforts reflect our steadfast dedication to upholding the utmost ethical standards in scientific exploration. Our objective is to protect the rights and privacy of all stakeholders involved, thereby fostering a culture of responsible and ethical research within our community.

## 8 REPRODUCIBILITY

Our FAST framework is implemented in PyTorch (Paszke et al., 2019). All the experiments are conducted on eight NVIDIA A100-80GB GPUs. Our full implementation shall be publicly released upon paper acceptance to guarantee reproducibility. The codes are available at the anonymous link *https://anonymous.4open.science/r/Sys2-LLaVA-8B0F/* for the review process.

All Experiments are conducted on eight NVIDIA A100-80GB SXM GPUs[1]. Reproducing the fine-tuning process would require approximately 15 A100 GPU days.

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

- §A provides **Implementation Details and Pseudo Code**.
- §B reports more **Results** for **Different Thinking Modes**.
- §C reports more **Quantitative Results** for **Visual Question Answering**.
- §D shows more **Quantitative Results** for **Segmentation**.
- §E analyzes **Failure Case**.
- §F examines the **Limitation and Future Work** of our research.
- §G discusses the **Social Impact** of our research.
- §H offers **Ethical Guard** or our dataset.
- §I claims **Reproducibility** of our approach.
- §J supplies **Data License** for the methods we used for comparison.

## A    IMPLEMENTATION DETAILS AND PSEUDO-CODE OF FAST

**Visual Resampler.** The resampler (Alayrac et al., 2022) compresses high-dimensional visual features into a fixed-size latent space using a cross-attention mechanism. It begins with a set of learnable latent embeddings, which query the vision encoder's output features through scaled dot-product attention. Each latent embedding attends selectively to the most relevant visual tokens, guided by attention weights computed via the query-key interaction. The process iterates across multiple layers of cross-attention, followed by feedforward transformations, refining the latent representations at each step. This approach ensures efficient dimensionality reduction while retaining critical information, producing a compact set of visual tokens for downstream tasks.

**Hyper-parameters.** We follow established methodologies and utilize LLaVA-v1.5 (Liu et al., 2024a) as the foundational visual agent. The image resolution is preprocessed to 336 × 336 pixels to accommodate the clip-vit-large-patch14-336 vision encoder (Radford et al., 2021). The AdamW optimizer (Loshchilov & Hutter, 2019) is employed with the DeepSpeed ZeRO 2 [2] configuration for fine-tuning the switch, proposal, and segmentation adapters with LoRA (Hu et al., 2022). For the LoRA configuration, we set the rank to 128 and alpha to 256, consistent with the settings of LLaVA-v1.5. Additionally, we adjust the learning rate of the vision encoder projection layer to 2e-5 to achieve better alignment. An MLP projection with channels of [256, 4096, 4096] is used to connect image representations into the word embedding space for the projection layer. An additional resampler projection layer is used to reduce the number of image tokens.

**Training Data for Switch Adapter.** Consistent with the pretraining stage of LLaVA-v1.5, we initially pretrain Vicuna-v1.5 as a base frozen large language model and for the MLP projection layer and sampler layer of the CLIP vision encoder using a 558K subset of the LAION-CC-SBU dataset [3] with BLIP (Li et al., 2022) captions. During the fine-tuning stage, we integrate the negative dataset acquired from $V^*$ (Wu & Xie, 2024) and PixelLM (Ren et al., 2024) and with the original LLaVA-v1.5 instruction tuning 665k data[4] for LoRA based finetuning.

Specifically, The dataset for fine-tuning the switch adapter was carefully constructed to emphasize scenarios requiring precise object recognition and complex reasoning. For the GQA subset of the 167k VQA data, we specifically targeted questions where the annotated objects mentioned in the query were critical for deriving the correct answer. Initially, the InstructBLIP model was used to evaluate GQA questions with annotated objects. Only questions that the model could correctly answer were retained. To ensure the importance of these annotated objects, we applied the LaMa image inpainting model to erase the mentioned objects from the corresponding images. The modified images were re-evaluated using InstructBLIP, and only questions that the model failed to answer

---

[2] https://github.com/microsoft/DeepSpeed
[3] https://huggingface.co/datasets/liuhaotian/LLaVA-Pretrain
[4] https://huggingface.co/datasets/liuhaotian/LLaVA-Instruct-150K

after object removal were included. This process ensured that the curated subset focused exclusively on questions where the annotated objects were essential, forming a robust component of the VQA data.

For the VAW object attribution dataset, both open-ended and binary questions about object attributes were synthesized. Open-ended questions were formulated around attributes such as "color," "material," and "pose," while binary questions incorporated additional attributes like "state" and "optical property." Answer formats adhered to predefined structures to ensure consistency. The same object removal and re-evaluation strategy as used in the GQA subset was applied, filtering the data to include only questions where the absence of objects rendered the query unanswerable.

From the LLaVA-80K instruction tuning data, noun phrases were extracted from the text of questions or instructions and matched with object category names defined by COCO, augmented with common synonyms such as "man" and "woman" for the "person" category. Images were retained only if the identified categories had annotated instances with bounding boxes. These annotated instances, along with their bounding box coordinates, were used as target objects during training.

In addition, we incorporated a data generation approach inspired by LISA, utilizing GPT-4 and GPT-4V to expand and diversify the dataset. Initially, LLAVA was used for image captioning, and GPT-4 generated questions about multiple regions in the image. While this approach utilized pre-existing mask annotations to reduce costs, its diversity was limited to the scope of the captions. To address these limitations, we refined the pipeline with GPT-4V, leveraging its advanced capabilities in visual understanding. Image captions, object names, and bounding box coordinates were input into GPT-4V, which, using dynamically crafted prompts, autonomously selected instances and generated nuanced question-answer pairs tailored to the image content. This refinement significantly improved the diversity and contextual relevance of the data. An illustrative example of such prompts is provided below:

```
Prompt: Imagine you need to query a machine agent about an
image. The image has a height of 720 pixels and a width of
1280 pixels. You are given several entities described by a
list, each identifying an object in the image along with its
location. The class names and corresponding coordinates are
as follows:
• Dog at [350.12, 450.45, 480.89, 600.67];
• Ball at [200.33, 300.22, 250.78, 350.56];
• Grass at [0.0, 500.0, 1280.0, 720.0];

Coordinates are represented as (top-left x, top-left
y, bottom-right x, bottom-right y). The question must
incorporate at least two of these objects and require
reasoning about the relationships or interactions between
them. Additional requirements for the generated question are
as follows:
1.The answer to the question must explicitly reference each
included object or its equivalent and avoid implying the
presence of any other objects not listed.
2.The question must be precise, meaningful, and avoid being
overly general.
3.The question should frame a single cohesive activity
or relationship rather than merely combining independent
sub-queries.
4.When answering, the class names should be rephrased to
indicate their position, role, or interaction in the image.
```

This multi-faceted dataset construction process ensured the generation of diverse and challenging samples, providing a robust foundation for fine-tuning the switch adapter on complex reasoning tasks.

**Training Data for Proposal Adapter** To determine the corresponding region for a query, we use LRP++ (Chefer et al., 2021) for data construction, similar to Chain of Spot (Liu et al., 2024b). Our initial prompt is as follows:

```
<Image>
To answer the question:  [Q],
where is the region of interest in the image based on [C]?

Ans.str[w₀, w₁, h₀, h₁]
```

The question $Q$ and the context clue $C$ are formatted to get the answer in terms of a bounding box. In this format, $w_0, w_1$ represent the left and right boundaries, respectively, while $h_0, h_1$ denote the upper and lower boundaries. To identify the correct region, we sampled one question per image from the LLaVA instruction tuning data, consisting of a total of 665k data for proposal finetuning.

**Training Data for Seg Adapter.** Adopting an approach similar to LISA Lai et al. (2024), the training data for our model comprises three distinct segments: a semantic segmentation dataset, a referring segmentation dataset, and a reasoning segmentation dataset. We deliberately exclude visual question-answering datasets to enhance the model's segmentation performance. The semantic segmentation segment includes the ADE20K (Zhou et al., 2017), COCO-Stuff (Caesar et al., 2018), and LVIS-PACO (Ramanathan et al., 2023) part segmentation datasets. The referring segmentation datasets encompass refCOCO Kazemzadeh et al. (2014), refCOCO+ (Kazemzadeh et al., 2014), refCOCOg (Caesar et al., 2018), and refCLEF (Rohrbach et al., 2016). The reasoning segmentation dataset includes ReasonSeg (Lai et al., 2024). It is important to note that the referring segmentation and reasoning segmentation datasets are carefully excluded during the evaluation of the segmentation benchmarks to prevent any potential data leakage.

**Pseudo-code Implementation.** The pseudo-code of FAST is given in Pseudo-code 1.

## B    MORE RESULTS FOR DIFFERENT THINKING MODES

As shown in Table 6, FaST demonstrates strong performance in both System 1 and System 2 reasoning on VQA datasets, outperforming baseline methods in most cases. Notably, FaST achieves the highest accuracy in challenging System 2 tasks across GQA, $VQA^T$, and $SQA^I$, which require advanced reasoning capabilities. This highlights the effectiveness of the switch adapter mechanism in dynamically allocating tasks based on complexity. While maintaining competitive performance in simpler System 1 tasks, FaST leverages its adaptive architecture to excel in more complex scenarios, as evidenced by its superior System 2 results.

Further, in Table 7, FaST's robustness extends to reasoning segmentation tasks, where it achieves significant improvements in System 2 performance compared to baseline models such as LLaVA with segmentation and LISA-7B. For example, in the ReasonSeg dataset, FaST records a remarkable 48.2 CIoU in System 2 tasks, significantly outperforming LISA-7B and LLaVA, which achieve 43.3 CIoU and 42.4 CIoU, respectively. This result underscores FaST's ability to generalize effectively across diverse task families and reasoning paradigms.

Overall, the results validate the universal applicability and robustness of the FaST framework. By effectively utilizing the switch adapter to allocate tasks dynamically, FaST demonstrates a strong capability to balance performance across both simple and complex reasoning tasks, making it a reliable solution for diverse real-world applications.

## C    MORE QUALITATIVE RESULTS FOR VISUAL QUESTION ANSWERING

Figure 7 presents additional qualitative results for Visual Question Answering (VQA). Our FAST framework consistently demonstrates remarkable performance across various challenging scenarios. Notably, in the bottom right corner of Figure 7, our FAST leverages extensive world knowledge to identify the keyboard, which subsequently aids in discovering the hidden computer mouse and providing the correct answer. This ability to integrate and utilize contextual information showcases the

**Algorithm 1:** Pseudo-code of FAST in a PyTorch-like style.

```python
class FaST:
    def __init__(self, switch_llm, proposal_llm, seg_llm):
        self.switch_llm = switch_llm
        self.proposal_llm = proposal_llm
        self.seg_llm = seg_llm

    def get_contextual_clues(self, image, question):
        # Get missing objects and context clues using switch adapter
        return self.switch_llm(image, question)

    # Construct Chain of Evidence
    def construct_coe(self, image, question, context_clues, missing_objects):
        # Step 1: Get region proposals
        region = self.proposal_llm(image, question, context_clues)

        # Step 2: Get pixel-level mask for the missing objects
        mask = self.seg_llm(region, missing_objects)

        return (context_clues, region, missing_objects, mask)

    # Main Function
    def forward(self, image, question):

        # Get initial answer
        initial_answer = self.switch_llm(image, question)

        # Check if slow thinking is needed based on the initial answer
        if "sorry, i can not answer" in initial_answer.lower():
            # Perform slow thinking
            missing_objects, context_clues = initial_answer['obj'], initial_answer
                ['clue']
            chain_of_evidence = self.construct_coe(image, question, context_clues,
                missing_objects)

            # Generate the final answer using the constructed chain of evidence
            final_answer = self.switch_llm(image, question, chain_of_evidence)
        else:
            # Perform fast thinking
            final_answer = initial_answer

        return final_answer
```

| Method | LLM | $VQA^{v2}$ | | GQA | | $VQA^T$ | | $SQA^I$ | |
|---|---|---|---|---|---|---|---|---|---|
| | | Sys 1 | Sys 2 | Sys 1 | Sys 2 | Sys 1 | Sys 2 | Sys 1 | Sys 2 |
| BLIP-2 | Vicuna-13B | 67.3 | 53.1 | 37.8 | 22.4 | 44.3 | 39.7 | 63.4 | 59.2 |
| LLaVA-v1.5 | Vicuna-7B | 81.2 | 68.0 | **70.3** | 47.0 | 61.1 | 53.7 | 68.4 | 65.7 |
| Chain of Spot | Vicuna-7B | **82.1** | 74.5 | 70.9 | 50.7 | **62.1** | 59.0 | **68.6** | 67.8 |
| FAST (Ours) | Vicuna-7B | 81.1 | **75.5** | 70.2 | **52.3** | 61.2 | **60.2** | 68.2 | **70.2** |

Table 6. **System 1 and System 2 performance on VQA datasets.** FaST demonstrates superior performance in both reasoning modes compared to baselines.

model's advanced capabilities and highlights its potential for practical applications. The qualitative results further underscore FAST's robustness and versatility in handling diverse VQA tasks.

## D MORE QUALITATIVE RESULTS FOR SEGMENTATION

Figure 8 showcases further qualitative results for the Segmentation task. Our FAST model excels in various challenging scenarios, accurately locating difficult targets and performing complex reasoning for more demanding queries. For instance, in the bottom right corner, the model successfully identifies an appliance that can be turned on when feeling hot by recognizing relevant contextual clues that suggest the appliance should probably appear on the wall, thereby resulting in the correct answer. This example demonstrates the model's advanced understanding, adaptability, and precision.

| Method | refCOCOg | | ReasonSeg | |
|---|---|---|---|---|
| | Sys 1 | Sys 2 | Sys 1 | Sys 2 |
| LISA-7B | 70.2 | 63.4 | 46.6 | 43.3 |
| LLaVA w Seg | 68.4 | 60.2 | 44.2 | 42.4 |
| FaST (Ours) | **70.8** | **64.1** | 46.4 | **48.2** |

Table 7. **System 1 and System 2 performance on reasoning segmentation tasks.** FaST achieves strong performance across both tasks, demonstrating its robustness and effectiveness in dynamic task allocation between System 1 and System 2.

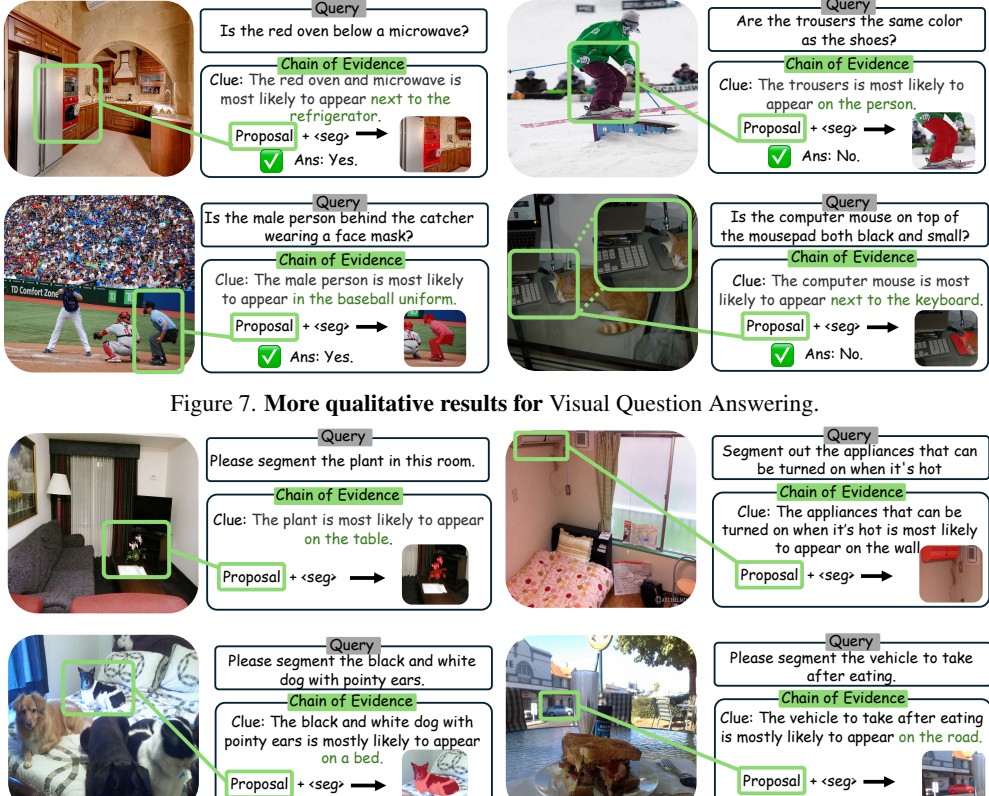

Figure 7. **More qualitative results for** Visual Question Answering.

Figure 8. **More qualitative results for** Visual Question Answering.

# E    FAILURE CASE

In Figure 9, we present an overview of the most notable failure cases, providing insights into the distinct patterns that lead to suboptimal outputs in our FAST model. These challenges include difficulty in triggering the *System 2* thinking mode, constructing adequate contextual clues, generating appropriate proposals, and providing accurate pixel masks. The model often fails to recognize the need for deliberate reasoning, relying instead on *System 1* thinking, which leads to incorrect responses, as seen in Figure 9a. Inadequate contextual clues generated by the switch adapter impair the model's focus on the correct region, resulting in vague or incorrect responses, as illustrated in Fig. 9b. The proposal adapter's inaccurate identification of regions of interest, as shown in Figure 9c, leads to proposals that do not correspond to the query. Additionally, the segmentation adapter struggles with producing precise masks, particularly for small or occluded objects, causing erroneous conclusions, as highlighted in Figure 9d. These failure cases underscore the urgent need for refinement in our FAST framework, emphasizing the importance of significantly enhancing the precision of the system switch adapter, improving contextual clue construction, and optimizing the proposal and segmentation adapters to achieve more reliable and consistently accurate responses in complex visual and textual tasks.

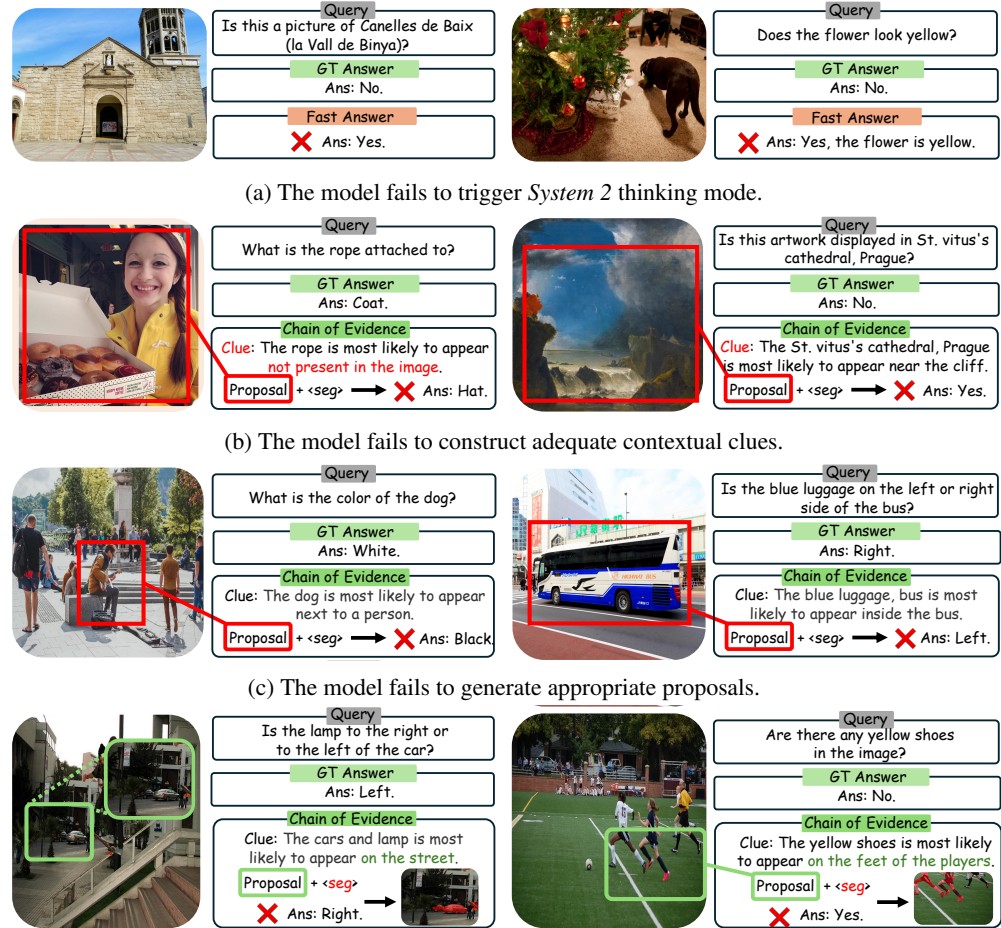

(a) The model fails to trigger *System 2* thinking mode.

(b) The model fails to construct adequate contextual clues.

(c) The model fails to generate appropriate proposals.

(d) The model fails to provide accurate pixel masks.

Figure 9. **Failure cases** of Our FAST system.

## F    LIMITATION AND FUTURE WORK

While the FAST framework has demonstrated significant advancements in emulating human-like cognitive processes in visual AI through its fast and slow thinking mechanisms, several limitations warrant attention. Firstly, the system's reliance on a predefined set of negative data for training the switch adapter may not encapsulate the full spectrum of real-world complexities. Firstly, this could lead to suboptimal performance when faced with novel or unexpected scenarios. Secondly, despite its fine-grained analysis capability, the pixel-level mask decoder might struggle with highly textured or patterned images where segmentation becomes challenging. Lastly, the generalizability of FAST across various domains and tasks necessitates further validation to ensure its robustness and reliability in diverse applications. We plan to develop advanced learning mechanisms that will allow the model to generalize more effectively beyond the predefined negative dataset. Additionally, we will focus on optimizing it for real-time applications to reduce computational overhead and response times.

For the recent large reasoning model like OpenAI o1 model, while these models leverage reinforcement learning and internal chains of thought to achieve scalability, they often require significant computational resources, making them less efficient.

In contrast, FaST is designed to prioritize multimodal reasoning with a clear focus on transparency and adaptability. The use of interpretable reasoning modes (System 1 and System 2) ensures that FaST provides insights into its decision-making processes, which is critical for applications requiring explainability. Additionally, FaST's modular design allows it to balance computational efficiency and accuracy dynamically, making it suitable for diverse and resource-constrained environments. These strengths highlight FaST's unique contributions and its complementary potential to scalable reasoning approaches.

## G    SOCIAL IMPACTS

The development and deployment of FAST significantly advance AI by enhancing the human-like cognitive abilities of LLM-based visual agents. Positively, FAST enables sophisticated applications in areas like vision-based dialogues and security surveillance, while its transparent decision-making fosters trust and ethical AI practices. However, reliance on large models and extensive datasets risks perpetuating biases, potentially leading to unjust or discriminatory outcomes. Addressing these ethical concerns and establishing responsible usage guidelines are essential for the responsible deployment of such advanced AI systems.

## H    ETHICAL SAFEGUARDS

In our paper introducing a novel framework FAST, we implement rigorous ethical measures to prevent potential misuse and promote responsible application. These measures are delineated in comprehensive protocols accompanying the final release of models and datasets. Our protocols encompass stringent usage guidelines, access controls, incorporation of safety filters, and monitoring systems. These concerted efforts reflect our steadfast dedication to upholding the utmost ethical standards in scientific exploration. Our objective is to protect the rights and privacy of all stakeholders involved, thereby fostering a culture of responsible and ethical research within our community.

## I    REPRODUCIBILITY

Our FAST framework is implemented in PyTorch (Paszke et al., 2019). All the experiments are conducted on eight NVIDIA A100-80GB GPUs. Our full implementation shall be publicly released upon paper acceptance to guarantee reproducibility. The codes are available at the anonymous link *https://anonymous.4open.science/r/Sys2-LLaVA-8B0F/* for the review process.

All Experiments (switch, proposal, and seg adapter) are conducted on eight NVIDIA A100-80GB SXM GPUs[5]. Reproducing the fine-tuning process would require approximately 15 A100 GPU days.

## J    LICENSES FOR EXISTING ASSETS

All the methods we used for comparison are publicly available for academic usage. The switch adapter is implemented based on the released code (https://github.com/penghao-wu/vstar) with an MIT license. The proposal adapter is implemented on the released code (https://github.com/dongyh20/Chain-of-Spot) with an Apache-2.0 license. The seg adapter is implemented on the released code (https://github.com/dvlab-research/LISA) with an Apache-2.0 license.

---

[5]https://www.nvidia.com/en-sg/data-center/a100/

