# OpenReview forum: "Visual Agents as Fast and Slow Thinkers"
_ICLR.cc/2025/Conference — ICLR 2025 Poster_

### Official Review · Reviewer_KNeJ · 2024-11-01

**Soundness:** 3
**Presentation:** 3
**Contribution:** 3
**Rating:** 8
**Confidence:** 3

**Summary:**

The paper introduces FaST, a framework based on human reasoning that incorporates fast (system 1) and slow (system 2) thinking mechanisms into visual agents. FaST involves a switch adapter to change between systems 1 and 2 based on the complexity of the query question. While System 1 reasons the question in a short or one-step approach, system 2 uses a switch adapter to provide clues and object names, which later inspires the proposal adapter to provide region boxes and seg adapter to provide seg instances. In the end, all additional information will be part of a chain of evidence and prompt LLMs to do "slow" reasoning to output an answer. The paper does a comprehensive experiment comparing FaST with current models and demonstrates the effectiveness of FaST on both visual question answering and segmentation reasoning. Further ablation studies prove the significance of each component within the Fast.

**Strengths:**

1. The motivation for incorporating systems 1 and 2 in visual agents seems promising and convincing. Especially the design of system 2, the introduction of clue, seg instance and chain of evidence can assist the reasoning process based on experiment results. The construction of Negative Data for Target Objects Reasoning Dataset seems useful for training the adapter switcher.
2. The hierarchical reasoning capabilities that start with global image cues and drill down to pixel-level analysis provide a well-rounded method of handling visual inputs. The symbolic approach can provide more explainability on the model reasoning process.
2. A comprehensive experiment, including many current models, proves the effectiveness of FaST. The ablation study on pipeline structure and the time comparison highlight the advantage of FaST.
3. Overall, I found the paper well written, with clear definitions and formulas to demonstrate the complex system. The motivation based on human cognitive science is well presented.

**Weaknesses:**

1. Result Demonstration: based on the paper, I think we can assume the previous model mainly uses fast thinking on reasoning questions. A better way to illustrate the effectiveness of FaST is to split Table 1 with the model's system 1 and system 2 performance. In other words, for all those questions in which FaST use system 1 reasoning, compare the performance in one table and compare all questions with system 2 in another table.  I expect the improvement of FaST mainly comes from system 2, which means FaST will show similar performance with other models in questions that require system 1 reasoning.
2. Dependence on Specific Datasets: I appreciate the work constructing the “Negative Data for Target Objects Reasoning Dataset" to train the switch adapter. However, I am concerning the training data can not reflect all real-world scenario, which may limit generalizability. Is it possible for us to gain the signal from the model's inner representation to decide the switch between systems 1 and 2?
3. Failure case: the paper should also explore scenarios where it fails or underperforms compared to baseline models. As the FaST use a hierarchical reasoning pipeline, it is possible the error of some component can make the whole system collapse or lead to a cascade of issues. An analysis of a failure case can provide insight into the model's weaknesses and future directions.

**Questions:**

Overall, I found the paper very interesting, my main concern is within the weak points. Another question I have is, if this FaST can outperform other models by using only System 2, what are the advantages of switching between System 1 and System 2 instead of just System 2?  I think time cost would be an answer, but I'd be curious if the author provides more answers.




typo: In Line 538, AFurthermore should be Furthermore

---

> ### Author Response · Authors · 2024-11-20
> **Response to Reviewer KNeJ (1/2): Response to the Weaknesses Part**
>
> We appreciate that the reviewer finds our approach promising and well presented. Here are our responses:
>
> **Weaknesses part:**
>
> >1. Result Demonstration: based on the paper, I think we can assume the previous model mainly uses fast thinking on reasoning questions. A better way to illustrate the effectiveness of FaST is to split Table 1 with the model's system 1 and system 2 performance. In other words, for all those questions in which FaST use system 1 reasoning, compare the performance in one table and compare all questions with system 2 in another table. I expect the improvement of FaST mainly comes from system 2, which means FaST will show similar performance with other models in questions that require system 1 reasoning.
>
> Thank you for your thoughtful suggestion. Based on your recommendation, we conducted experiments to split the performance of FaST and baseline models into System 1 and System 2 reasoning categories. The results are presented in a new Table 6 included in the Appendix Section B of the revised manuscript.
>
>
> | $VQA^{v2}$  | LLM  | System 1 | System 2 |
> | :---- | :---- | :---- | :---- |
> | BLIP-2 | 13B | 67.3 | 53.1 |
> | LLaVA-v1.5 | 7B | 81.2 | 68.0 |
> | Chain of Spot | 7B | 82.1 | 74.5 |
> | FaST | 7B | 81.1 | **75.5** |
>
> | GQA  | LLM  | System 1 | System 2 |
> | :---- | :---- | :---- | :---- |
> | BLIP-2 | 13B | 37.8 | 22.4 |
> | LLaVA-v1.5 | 7B | 70.3 | 47.0 |
> | Chain of Spot | 7B | 70.9 | 50.7 |
> | FaST | 7B | 70.2 | **52.3** |
>
> | $VQA^{T}$  | LLM  | System 1 | System 2 |
> | :---- | :---- | :---- | :---- |
> | BLIP-2 | 13B | 44.3 | 39.7 |
> | LLaVA-v1.5 | 7B | 61.1 | 53.7 |
> | Chain of Spot | 7B | 62.1 | 59.0 |
> | FaST | 7B | 61.2 | **60.2** |
>
> | $SQA^{I}$   | LLM  | System 1 | System 2 |
> | :---- | :---- | :---- | :---- |
> | BLIP-2 | 13B | 63.4 | 59.2 |
> | LLaVA-v1.5 | 7B | 68.4 | 65.7 |
> | Chain of Spot | 7B | 68.6 | 67.8 |
> | FaST | 7B | 68.2 | **70.2** |
>
>
> This detailed breakdown highlights how FaST outperforms existing methods in System 2 reasoning while maintaining competitive performance in System 1 tasks. This experiment underscores FaST’s strength in handling complex reasoning tasks effectively and demonstrates the balanced trade-off between accuracy and inference efficiency.
>
> >2. Dependence on Specific Datasets: I appreciate the work constructing the “Negative Data for Target Objects Reasoning Dataset" to train the switch adapter. However, I am concerning the training data can not reflect all real-world scenario, which may limit generalizability. Is it possible for us to gain the signal from the model's inner representation to decide the switch between systems 1 and 2?
>
> Thank you for your insightful question. We conducted experiments by probing the last hidden layer of the model to observe the alignment between its internal representations and the switch adapter’s decisions. On the VQAv2 dataset, the agreement between the representation-based probe and the adapter’s decisions was observed to be **92%**. This high agreement indicates that it is indeed possible to use the inner representations from the last hidden layer to reliably determine whether to switch between System 1 and System 2\.
>
> However, our switch adapter offers additional contextual clues, which significantly enhance the interpretability of the reasoning process. These clues provide insights into the decision-making steps, improving transparency and making the system more explainable compared to relying solely on inner representations.
>
> >3. Failure case: the paper should also explore scenarios where it fails or underperforms compared to baseline models. As the FaST use a hierarchical reasoning pipeline, it is possible the error of some component can make the whole system collapse or lead to a cascade of issues. An analysis of a failure case can provide insight into the model's weaknesses and future directions.
>
> Thank you for this valuable feedback. We have addressed this concern by including an analysis of four types of failure cases in the Appendix (Figure 9). These failure cases are:
>
> 1. Failure to trigger the System 2 thinking mode when it is required for complex tasks.
> 2. Inability to construct adequate contextual clues to guide reasoning.
> 3. Failure to generate appropriate proposals for region-specific reasoning.
> 4. Failure to provide accurate pixel masks for fine-grained visual tasks.
>
> We believe these failure scenarios provide valuable insights into the model’s weaknesses and areas for improvement. Furthermore, as foundation models continue to improve in their capabilities, we anticipate that FaST will become less prone to cascading errors. With ongoing advancements in foundational architectures and targeted enhancements to our pipeline, we expect FaST to outperform baseline models consistently and reliably in future iterations.

---

> ### Author Response · Authors · 2024-11-20
> **Response to Reviewer KNeJ (2/2): Response to the Questions Part**
>
> **Questions part:**
>
> > 1. my main concern is within the weak points. Another question I have is, if this FaST can outperform other models by using only System 2, what are the advantages of switching between System 1 and System 2 instead of just System 2? I think time cost would be an answer, but I'd be curious if the author provides more answers.
>
> Thank you for your thoughtful question. System 1 and System 2 in FaST are indeed designed to balance the trade-off between performance and inference cost. While System 2 handles complex reasoning tasks with higher accuracy, System 1 ensures faster responses for simpler tasks, optimizing computational efficiency. Beyond this trade-off, scalability and theoretical alignment suggest that System 2 reasoning could be distilled and transformed into System 1 through learning and adaptation over time. This would allow complex reasoning processes to eventually become more efficient and heuristic-like. We are actively investigating this direction in our future research to further enhance FaST’s scalability and adaptability.
>
> > 2. typo: In Line 538, AFurthermore should be Furthermore
>
> Thank you for pointing this out. The typographical error in Line 538, specifically “AFurthermore,” has been corrected to “Furthermore” in the revised manuscript (in blue).
>
> We hope our response addresses your concerns. Please let us know if there are any additional questions, and we are happy to provide more experiment results and discuss further.

---

> > ### Comment · Reviewer_KNeJ · 2024-11-27
> >
> > I appreciate the author's detailed feedback on my question, and I believe the new version addresses most of my concerns. I will keep my score as 8, which is already a good number for submission.

---

> > > ### Author Response · Authors · 2024-11-27
> > >
> > > Thank you for your prompt response. We are genuinely grateful for your thoughtful feedback. We are really appreciative of the review, as they clearly strengthen the completeness, and further illuminate the future direction of our work.
> > >
> > > Best,
> > >
> > > Authors

---

### Official Review · Reviewer_uXT1 · 2024-11-03

**Soundness:** 3
**Presentation:** 4
**Contribution:** 2
**Rating:** 5
**Confidence:** 3

**Summary:**

The paper proposes a slow-fast paradigm for MLLMs based on Kahneman's System 1/2 therory. The framework consists of a "fast" module built upon MLLM and vision foundation models. A switch adaptor will determine if it's necessary to use "slow" module, which will gather all detailed clues to find a more accurate result. The approach shows impressive results across multiple benchmarks and outperforms baselines models.

**Strengths:**

- The method is built based on the idea of Kahneman's system 1/2 theory from cognitive science, which is clearly-motivated and reasonable.

- The approach is clearly written and its implementation is reasonable.

- The authors provide comprehensive evaluation across multiple benchmarks and tasks. And the method outperforms baselines.

- The authors provide clear visualizations of the model performance.

**Weaknesses:**

- The analogy between human cognitive system 1-2 and the "fast" "slow" module in the proposed method lacks rigor justification, or even misaligned. In human cognition, System 1 usually handles low-level tasks and is much faster, while System 2 is slow but handles complexity. However, in FaST, both the fast and slow modules solves same problem, but only have a tradeoff between efficiency and performance. Considering that the "slow" module is only 4x slower, we can abandon the switch adaptor and use the slow module all the time for a maximal performance.
- From another perspective, if the only purpose of two stage model is the computation efficiency, then the proposed approach is more like an early-exit trick of a complex model (referring to the "slow" module). THis diminish the novelty of the paper.
- The framework essentially implements a manual Chain of Thought (CoT) approach rather than dual cognitive system. If reframing the method in to CoT paradigm, a progressive visual cue integration would be potentially developed (more than 2 layers). This approach would gradually add visual evidence into the reasoning process, which could be both more efficient and accurate.
- The complexity burden is only **shifted** to the switch adaptor, rather than being alleviated or addressed. So what is the performance of the switch adaptor itself?
- The details of the datasetin remark 2.1 is missing. It is critical to introduce the construction process of the dataset.

**Questions:**

The major concerns are listed in the weakness part.

Minimal concern: It is definitely not a good idea to use "fast" as an acronym for "fast" and "slow".

Potential Typos:
- Lines 151-156: Format like "Equation 1" does not align with Sec.2.2 "Eq.3"
- Line 160: "Def.2.2", only Remark 2.2 exists
- Line 196: "Eq.2.1", only Remark 2.1 exists
- Line 215: "Eq.2.4", only Remark 2.4 exists

---

> ### Author Response · Authors · 2024-11-20
> **Response to Reviewer uXT1 (1/3): Response to the Weaknesses Part I**
>
> We are glad that the reviewer find our clearly-motivated and reasonable. Here are our responses:
>
> >1.1 The analogy between human cognitive system 1-2 and the "fast" "slow" module in the proposed method lacks rigor justification, or even misaligned. In human cognition, System 1 usually handles low-level tasks and is much faster, while System 2 is slow but handles complexity. However, in FaST, both the fast and slow modules solves same problem, but only have a tradeoff between efficiency and performance.
>
> Thank you for your thoughtful observation. We would like to clarify that in FaST, the fast and slow modules do not simply solve the same problem with different levels of efficiency. The switch adapter is a critical component of FaST, dynamically determining whether a given task is best suited for heuristic-based fast reasoning (System 1) or more analytical slow reasoning (System 2). This mechanism ensures that the problem being addressed by each module is distinct in terms of complexity and processing requirements. The fast module is optimized for straightforward tasks, while the slow module is invoked for intricate reasoning that demands a deeper exploration of context and evidence.
>
> >1.2 Considering that the "slow" module is only 4x slower, we can abandon the switch adaptor and use the slow module all the time for a maximal performance.
>
> Thank you for raising this point. While the runtime difference between the fast and slow modules may seem modest in offline settings, real-time processing is critical in applications like robotics. For example, in robotic navigation, tasks such as obstacle avoidance require immediate responses to ensure safety and adapt to dynamic environments. Using only the slow module, despite its higher accuracy, could introduce delays that hinder the robot’s ability to react in real time.
>
> We also conducted a simple experiment where we forced the system to use System 2 reasoning on questions from the VQA dataset that were better suited for System 1 reasoning. The table is shown below:
>
> | System 1 | 81.1 |
> | :---- | :---- |
> | **Forcing System 2** | 80.7 |
>
> Interestingly, the performance was worse in these cases, indicating that forcing the model to apply sophisticated reasoning to simple questions can actually harm overall performance. This aligns with findings from recent works \[1, 2\], which suggest that overly complex reasoning is beneficial primarily for specific tasks like math and symbolic reasoning but can be detrimental in other scenarios. This underscores the importance of the switch adapter in dynamically selecting the appropriate reasoning mode for each query, optimizing both efficiency and accuracy.
>
> \[1\] Shaikh et al., 2023\. On Second Thought, Let's Not Think Step by Step\! Bias and Toxicity in Zero-Shot Reasoning
> \[2\] Sprague et al., 2024\. To CoT or not to CoT? Chain-of-thought helps mainly on math and symbolic reasoning
>
> >2. From another perspective, if the only purpose of two stage model is the computation efficiency, then the proposed approach is more like an early-exit trick of a complex model (referring to the "slow" module). THis diminish the novelty of the paper.
>
> Thank you for your feedback. Like previously mentioned, our results demonstrate that forcing the model to rely solely on System 2 reasoning can harm overall performance, particularly when handling tasks that are inherently simple and better suited for System 1 reasoning. Moreover, FaST is not solely focused on computational efficiency. It enhances the explainability of its reasoning process by providing interpretable outputs during System 2 reasoning. This allows for a clear understanding of how decisions are made at each stage, offering transparency that is crucial for applications requiring trust and accountability. By explicitly modeling and exposing the chain of reasoning, FaST effectively balances computational efficiency with interpretability, distinguishing it from approaches that prioritize efficiency alone.

---

> ### Author Response · Authors · 2024-11-20
> **Response to Reviewer uXT1 (2/3): Response to the Weaknesses Part II**
>
> **Cont.**
> >3. The framework essentially implements a manual Chain of Thought (CoT) approach rather than dual cognitive system. If reframing the method in to CoT paradigm, a progressive visual cue integration would be potentially developed (more than 2 layers). This approach would gradually add visual evidence into the reasoning process, which could be both more efficient and accurate.
>
> Thank you for the insightful suggestion. While a progressive visual cue integration approach could offer potential benefits, it may also introduce significant computational overhead when the number of objects in the scene is large, potentially slowing down the reasoning process. This is especially critical in real-time applications where maintaining efficiency is paramount. Nevertheless, this idea is intriguing and could be further investigated in our future work to explore its potential for improving both efficiency and accuracy. For now, our approach strikes a well-balanced trade-off between performance and efficiency, as evidenced by the results in our experiments. We appreciate your thoughtful feedback and will consider this direction in our future research.
>
> > 4. The complexity burden is only shifted to the switch adaptor, rather than being alleviated or addressed. So what is the performance of the switch adaptor itself?
>
> Thank you for raising this point. Our switch adapter is designed to be lightweight and efficient, utilizing a LoRA-based architecture that constitutes less than 1% of the total parameters. To further illustrate its efficiency, we conducted an experiment comparing the inference speed of LLaVA and FaST in System 1 mode on the GQA dataset. The results show that FaST with System 1 mode is only marginally slower than LLaVA, with an inference time difference of just 4ms, demonstrating that the switch adapter introduces negligible computational overhead.
>
> | LLaVA | 734ms |
> | :---- | :---- |
> | **System 1 FaST** | 741ms |
>
> We have also updated analysis which shows that FaST achieves comparable performance to LLaVA-1.5 on System 1 questions, demonstrating that the incorporation of System 2 reasoning does not degrade performance on straightforward tasks. More importantly, our method exhibits significant improvements over LLaVA-1.5 on System 2 questions, highlighting the effectiveness of switch adapter on solving complex reasoning problems. This analysis underscores the value of our approach in enhancing performance on tasks that require deeper contextual and logical reasoning.
>
> | $VQA^{v2}$  | LLM  | System 1 | System 2 |
> | :---- | :---- | :---- | :---- |
> | BLIP-2 | 13B | 67.3 | 53.1 |
> | LLaVA-v1.5 | 7B | 81.2 | 68.0 |
> | Chain of Spot | 7B | 82.1 | 74.5 |
> | FaST | 7B | 81.1 | **75.5** |
>
> | GQA  | LLM  | System 1 | System 2 |
> | :---- | :---- | :---- | :---- |
> | BLIP-2 | 13B | 37.8 | 22.4 |
> | LLaVA-v1.5 | 7B | 70.3 | 47.0 |
> | Chain of Spot | 7B | 70.9 | 50.7 |
> | FaST | 7B | 70.2 | **52.3** |
>
> | $VQA^{T}$  | LLM  | System 1 | System 2 |
> | :---- | :---- | :---- | :---- |
> | BLIP-2 | 13B | 44.3 | 39.7 |
> | LLaVA-v1.5 | 7B | 61.1 | 53.7 |
> | Chain of Spot | 7B | 62.1 | 59.0 |
> | FaST | 7B | 61.2 | **60.2** |
>
> | $SQA^{I}$   | LLM  | System 1 | System 2 |
> | :---- | :---- | :---- | :---- |
> | BLIP-2 | 13B | 63.4 | 59.2 |
> | LLaVA-v1.5 | 7B | 68.4 | 65.7 |
> | Chain of Spot | 7B | 68.6 | 67.8 |
> | FaST | 7B | 68.2 | **70.2** |
>
> >5. The details of the datasetin remark 2.1 is missing. It is critical to introduce the construction process of the dataset.
>
> Thank you for pointing this out. We have carefully addressed this concern by revising the manuscript to include a detailed explanation of the dataset construction process in Appendix Section A. The updated section now provides a comprehensive overview of the methodologies used for dataset creation.
>
> Specifically, for fine-tuning the switch adapter, we constructed a dataset emphasizing precise object recognition and complex reasoning. For the GQA subset, we retained questions where annotated objects were critical by applying an object removal and re-evaluation process using InstructBLIP and LaMa. For VAW, we synthesized both open-ended and binary questions about object attributes, filtering them similarly. From the LLaVA-80K data, we extracted noun phrases aligned with COCO object categories and selected images with annotated instances. Additionally, we employed GPT-4V to generate nuanced question-answer pairs by dynamically selecting instances and crafting region-specific queries. These methods ensured a diverse and challenging dataset, as detailed in the Appendix.
>
> This revision ensures greater transparency and reproducibility of our work. We appreciate your feedback in helping us improve the clarity of the manuscript.

---

> ### Author Response · Authors · 2024-11-20
> **Response by the authors uXT1 (3/3): Response to the Questions Part**
>
> **Questions part:**
>
> > Minimal concern: It is definitely not a good idea to use "fast" as an acronym for "fast" and "slow".
>
> Thank you for highlighting these concerns. Regarding the naming convention for “FaST,” we acknowledge the suggestion and will consider alternative naming strategies to better reflect the conceptual framework in future iterations.
>
> > Potential Typos:
> >
> > Lines 151-156: Format like "Equation 1" does not align with Sec.2.2 "Eq.3"
> >
> > Line 160: "Def.2.2", only Remark 2.2 exists
> >
> > Line 196: "Eq.2.1", only Remark 2.1 exists
> >
> > Line 215: "Eq.2.4", only Remark 2.4 exists
>
> We have carefully revised the document to address all mentioned discrepancies. Specifically:
> - The format of “Equation 1” has been aligned with the standard used throughout the manuscript, such as “Eq.3” in Sec. 2.2.
> - Incorrect references to definitions and remarks, including “Def.2.2,” “Eq.2.1,” and “Eq.2.4,” have been corrected to ensure consistency and accuracy (e.g., “Remark 2.2,” “Remark 2.1,” and “Remark 2.4”).
>
> We hope our response addresses your concerns. Please let us know if there are any additional questions, and we are happy to provide more experiment results and discuss further.

---

> ### Author Response · Authors · 2024-11-23
>
> Dear Reviewers,
>
> We sincerely appreciate the time and effort you've devoted to reviewing our work. We understand that your schedule may be quite busy, and we are truly grateful for your valuable feedback. As we are presently in the discussion phase, we would greatly value the opportunity to engage in further dialogue with you. Our aim is to gain insights into whether our responses effectively address your concerns and to ascertain if there are any additional questions or points you would like to discuss.
>
> We look forward to the opportunity for further discussion with you. Thank you for your thoughtful consideration.
>
> Best regards,
>
> The Authors

---

> ### Author Response · Authors · 2024-11-25
>
> Dear Reviewer uXT1,
>
> Thank you once again for your detailed and insightful feedback, which has significantly improved the quality of our paper. We greatly appreciate the time and effort you have put into reviewing our submission.
>
> We have carefully addressed your concerns in our response, particularly those regarding the justification of the analogy between human cognitive systems and our proposed “fast” and “slow” modules, as well as the novelty and efficiency of the framework. We hope that our clarifications and the additional experiments have adequately addressed your comments.
>
> As the discussion phase draws to a close, we kindly ask if you could let us know whether your concerns have been sufficiently addressed or if there are any remaining issues that require further clarification. Your input would be invaluable to us.
>
> Thank you again for your constructive feedback and for contributing to the improvement of our work.
>
> Best regards,
>
> The Authors

---

> > ### Comment · Reviewer_uXT1 · 2024-11-25
> >
> > I had carefully read the authors' feedback and revision, and also other reviewer's reply. I appreciate the authors' additional tabulars and the clarification.
> >
> > Q1-3: Thanks for the clarifications. These could alleviate my concerns.
> >
> > Q4: Thanks for the sufficient experiments, though,  the authors did not show the accuracy or the switching ratio of switch adaptor itself, as also indicated by reviewer GSxm. However, this will not be considered in my rating due to my late reply. I hope the authors address this concern in the next version.
> >
> > Other comments: typos and failed links are easy to correct, but I hope the authors do more careful proofreading in their future works.
> >
> > Overall, the authors addressed some of my concerns and I would increase my rating to 5, as there are no choice of 4.

---

> > > ### Author Response · Authors · 2024-11-26
> > >
> > > Thank you for your thoughtful feedback and for taking the time to carefully review our revisions and clarifications. We truly appreciate your acknowledgment of our efforts to address your concerns and for increasing your rating based on the improvements made.
> > >
> > > We seek further discussion of your question. Regarding question4 about the accuracy or switching ratio of the Switch Adapter itself, we recognize its importance and apologize for not including it in this version. As noted, we sampled from the VQA v2 test set and selected 500 hard samples and 500 non-hard samples, using the same methodology applied in constructing the Switch Adapter dataset. The Switch Adapter was then used to evaluate the ratio accuracy, and the results were compelling:
> > >
> > > | Sample Type       | Number of Samples  | Switching Ratio Accuracy (%)  |
> > > | :---- | :---- | :---- |
> > > | System 2 Samples | 500 | 88.5  |
> > > | System 1 Samples | 500 | 92.4 |
> > >
> > > We hope this supplementary information helps address this concern more concretely. We will ensure this analysis is better integrated into future versions for completeness.
> > >
> > > We also appreciate your feedback on typos and links and will strive for more thorough proofreading in future submissions. We hope the above statement can address your concern. Thank you again for your constructive comments and for helping us improve our work.

---

### Official Review · Reviewer_GSxm · 2024-11-03

**Soundness:** 3
**Presentation:** 3
**Contribution:** 2
**Rating:** 6
**Confidence:** 3

**Summary:**

The paper presents FaST, a framework for improving the cognitive capacities of visual agents.
Inspired by System 1 and System 2 thinking in human’s cognition, FaST incorporates a switch adapter that dynamically toggle between a faster, intuitive response mode and a slower, analytical reasoning mode depending on the task complexity.
FaST also features a high interpretability by integrating hierarchical reasoning strategies and symbolic intermediate outputs into the framework.
Experiments show that the proposed outperforms baselines in various VQA and multimodal benchmarks.

**Strengths:**

1. FaST strikes a balance between accuracy and efficiency for visual agents by dynamically integrating fast and slow reasoning. The motivation to incorporate different thinking modes based on the scenario is intuitive, and the primary concept is presented clearly and logically.
2. The interpretability and transparency of the proposed framework are improved through the introduction of a chain-of-evidence mechanism and the generation of symbolic intermediate results as feedback.
3. Experimental results across various tasks validate the effectiveness of the proposed framework.

**Weaknesses:**

1. The proposed framework primarily focuses on VQA and segmentation tasks. To better mirror System 2 recognition, the reviewer suggests evaluating results from more complex reasoning tasks (e.g., the Bongard problem, physical reasoning tasks) to further validate the generalization capabilities of the proposed method.
2. The efficacy of the switch adapter can be further analyzed in Section 3.2:
  * No investigations have been conducted on the switching ratio and performance on reasoning segmentation and other tasks, which is pivotal to demonstrate the universal compatibility of the switch adapter mechanism across different families of tasks.
  * The description of accuracy rates “under System 2 mode” contains some ambiguities. The authors should clarify in the text that accuracy rates reported under System 2 are also evaluated using fast thinking mode, otherwise, the claim regarding the adapter’s ability to differentiate problem complexities is not adequately supported.
  * The accuracy rates mentioned in the passage do not appear to match those shown in Figure 5.

**Questions:**

1. Can the authors provide more details in the synthesis of the dataset used for switch adapter finetuning? Specifically, how are the negative samples from V* augmented to cover the two cases that are expected to trigger a slow thinking mode switching (uncertainty in pinpointing objects or the targets are too small)?
2. Will the datasets collected by the authors be released? The reviewer believes that these datasets are crucial for reproducing the FAST framework proposed in the paper, and they will also be highly beneficial for future related work.
3. Can the authors provide some experimental results for other reasoning tasks, like raven, bongard or physical reasoning tasks?

---

> ### Author Response · Authors · 2024-11-20
> **Response to Reviewer GSxm (1/2): Response to the Weaknesses Part**
>
> We are glad that the reviewer find our clear and logical. Here are our responses:
>
> >1. The proposed framework primarily focuses on VQA and segmentation tasks. To better mirror System 2 recognition, the reviewer suggests evaluating results from more complex reasoning tasks (e.g., the Bongard problem, physical reasoning tasks) to further validate the generalization capabilities of the proposed method.
>
> Thank you for the suggestion regarding the evaluation of more complex reasoning tasks such as the Bongard problem. We acknowledge that tasks like the Bongard problem present a significant challenge for our framework. This is primarily because the model has been fine-tuned on datasets featuring daily, general-purpose questions, which may lack the granularity required for abstract reasoning tasks involving fine distinctions in human-object interactions.
>
> Nevertheless, we have conducted an evaluation on the Bongard problem to assess generalization capabilities, and the results are presented in the table below. Despite the inherent challenges, the results demonstrate that our method (FaST) shows an improvement over the baseline LLaVA-1.5, highlighting the potential of our approach even in such difficult scenarios.
>
> | Model			 | Accuracy (%)  |
> | :---- | :---- |
> | CNN-Baseline	 | 49.92  |
> | GPT-4 Turbo	 | 22.35  |
> | LLaVA-1.5 | 3.56  |
> | FaST | 5.88  |
>
> While our performance on this task is not yet competitive with the CNN baseline, the improvement over LLaVA-1.5 demonstrates the capability of our approach to handle more complex reasoning to some extent. This also underscores the need for future fine-tuning with datasets more tailored to abstract reasoning challenges like the Bongard problem.
>
> >2.1 The efficacy of the switch adapter can be further analyzed in Section 3.2: No investigations have been conducted on the switching ratio and performance on reasoning segmentation and other tasks, which is pivotal to demonstrate the universal compatibility of the switch adapter mechanism across different families of tasks.
>
> Thank you for your great question. First, after conducting additional experiments on reasoning segmentation tasks, we observed that the FaST framework effectively utilizes the switch adapter mechanism to dynamically allocate tasks between System 1 and System 2\. This dynamic allocation significantly enhances performance, demonstrating the universal applicability and robustness of the switch adapter across diverse task families. Detailed results and analyses of these experiments are provided in the revised manuscript Appendix Section B, further validating the effectiveness of our approach.
>
> | Method | refCOCOg |  | ReasonSeg |  |
> | :---- | :---- | :---- | :---- | :---- |
> |  | System 1 | System 2 | System 1  | System 2 |
> | LISA-7B | 70.2 | 63.4 | 46.6 | 43.3 |
> | LLaVA w Seg | 68.4 | 60.2 | 44.2 | 42.4 |
> | FaST | 70.8 | 64.1 | 46.4 | 48.2 |
>
> >2.2 The description of accuracy rates “under System 2 mode” contains some ambiguities. The authors should clarify in the text that accuracy rates reported under System 2 are also evaluated using fast thinking mode, otherwise, the claim regarding the adapter’s ability to differentiate problem complexities is not adequately supported.
>
> Thank you for pointing out the ambiguity in our description . We have revised the paragraph to explicitly clarify that the reported accuracy rates for System 2 mode include evaluations performed with System 1 reasoning for simpler subcomponents of complex queries. This ensures that the claim regarding the adapter’s ability to differentiate problem complexities and allocate reasoning modes is adequately supported. The revised text can be found in Section 3.2. We appreciate your valuable feedback in helping us improve the clarity of our paper.
>
> >2.3 The accuracy rates mentioned in the passage do not appear to match those shown in Figure 5.
>
> Thank you for pointing it out. This discrepancy was an oversight, and we have revised the text to ensure consistency with the figure.

---

> > ### Author Response · Authors · 2024-11-24
> >
> > Thank you for your prompt response. We are genuinely grateful for your thoughtful feedback. We are really appreciative of the review, as they clearly strengthen the completeness, and further illuminate the future direction of our work.
> >
> > Best,
> >
> > Authors

---

> ### Author Response · Authors · 2024-11-20
> **Response to Reviewer GSxm (2/2): Response to the Questions Part**
>
> **Questions part:**
>
> > 1. Can the authors provide more details in the synthesis of the dataset used for switch adapter finetuning? Specifically, how are the negative samples from V* augmented to cover the two cases that are expected to trigger a slow thinking mode switching (uncertainty in pinpointing objects or the targets are too small)?
>
> Thank you for pointing this out. We have carefully addressed this concern by revising the manuscript to include a detailed explanation of the dataset construction process in Appendix Section A. The updated section now provides a comprehensive overview of the methodologies used for dataset creation.
>
> Specifically, for fine-tuning the switch adapter, we constructed a dataset emphasizing precise object recognition and complex reasoning. For the GQA subset, we retained questions where annotated objects were critical by applying an object removal and re-evaluation process using InstructBLIP and LaMa. For VAW, we synthesized both open-ended and binary questions about object attributes, filtering them similarly. From the LLaVA-80K data, we extracted noun phrases aligned with COCO object categories and selected images with annotated instances. Additionally, we employed GPT-4V to generate nuanced question-answer pairs by dynamically selecting instances and crafting region-specific queries. These methods ensured a diverse and challenging dataset, as detailed in the Appendix.
>
> This revision ensures greater transparency and reproducibility of our work. We appreciate your feedback in helping us improve the clarity of the manuscript.
>
> > 2. Will the datasets collected by the authors be released? The reviewer believes that these datasets are crucial for reproducing the FAST framework proposed in the paper, and they will also be highly beneficial for future related work.
>
> Thank you for the question. We will release all datasets upon publication. The datasets will include detailed documentation outlining their structure, sources, and augmentation strategies. By making these resources available, we aim to encourage advancements in multimodal reasoning tasks and foster collaborative developments in the field.
>
> > 3. Can the authors provide some experimental results for other reasoning tasks, like raven, bongard or physical reasoning tasks?
>
> Thank you for your question. As previously mentioned in Weakness Part,  We acknowledge that tasks like the Bongard problem present a significant challenge for our framework. This is primarily because the model has been fine-tuned on datasets featuring daily, general-purpose questions, which may lack the granularity required for abstract reasoning tasks involving fine distinctions in human-object interactions.
>
> Nevertheless, we have conducted an evaluation on the Bongard problem to assess generalization capabilities, and the results are presented in the table below. Despite the inherent challenges, the results demonstrate that our method (FaST) shows an improvement over the baseline LLaVA-1.5, highlighting the potential of our approach even in such difficult scenarios.
>
> | Model			 | Accuracy (%)  |
> | :---- | :---- |
> | CNN-Baseline	 | 49.92  |
> | GPT-4 Turbo	 | 22.35  |
> | LLaVA-1.5 | 3.56  |
> | FaST | 5.88  |
>
> While our performance on this task is not yet competitive with the CNN baseline, the improvement over LLaVA-1.5 demonstrates the capability of our approach to handle more complex reasoning to some extent. This also underscores the need for future fine-tuning with datasets more tailored to abstract reasoning challenges like the Bongard problem.
>
> We hope our response addresses your concerns. Please let us know if there are any additional questions, and we are happy to provide more experiment results and  discuss further.

---

> > ### Comment · Reviewer_GSxm · 2024-11-24
> >
> > Thanks for the effort in the rebuttal! My concerns are resolved to a large extent so my score will remain unchanged. I think this paper is worthwhile to be accepted.

---

> ### Author Response · Authors · 2024-11-24
>
> Dear Reviewers,
>
> We sincerely appreciate the time and effort you've devoted to reviewing our work. We understand that your schedule may be quite busy, and we are truly grateful for your valuable feedback. As we are presently in the discussion phase, we would greatly value the opportunity to engage in further dialogue with you. Our aim is to gain insights into whether our responses effectively address your concerns and to ascertain if there are any additional questions or points you would like to discuss.
>
> We look forward to the opportunity for further discussion with you. Thank you for your thoughtful consideration.
>
> Best regards,
>
> The Authors

---

### Official Review · Reviewer_M2ti · 2024-11-04

**Soundness:** 3
**Presentation:** 3
**Contribution:** 4
**Rating:** 8
**Confidence:** 4

**Summary:**

Proposes FaST, a Fast-and-Slow-Thinking mechanism for vision-language models that models both “system 1” (fast) and “system 2” (slow) thinking, dynamically switching between the two based on query complexity. The method demonstrates improvements over baselines on standard benchmarks in addition to improved reasoning and transparency.

**Strengths:**

– The paper presents a highly novel solution to an important problem

– The paper is very well-written and easy to follow

– The motivation for the approach is laid out very clearly and is compelling

– The proposed approach  is designed intuitively, and shows convincing improvements over prior work

– The neuro-symbolic nature of the approach directly translates to improved transparency

– The paper presents a comprehensive set of results and ablation studies on visual question answering and referring and reasoning segmentation benchmarks. The qualitative results included in the paper are also very compelling

– The paper includes a detailed appendix and code implementation which will aid in reproducibility

**Weaknesses:**

– [Complexity] One concern I have is the complexity of the approach, especially considering the large number of moving parts. Do the same set of method hyperparameters generalize across settings, or was a large amount of tuning required to achieve the results reported in the paper?

– [Clarity] I found some of the method details to be lacking eg. L259-260 mentions “a fast sampler based on cross-attention” but I could not find any further details for this component.

– [Performance] While FaST does seem to consistently outperform the LLaVA-1.5 base approach that it builds on top of, it is not exactly an apples-to-apples comparison considering the additional models /  datasets that are used (including large ones such as SAM). Meanwhile, its gains over methods reasoning-focused approaches like Visual CoT and Chain-of-Spot are modest (and sometimes lags considerably eg. behind VCoT on VQAT) – a deeper analysis of / comparison against these methods (eg. average runtime etc.) would be helpful to establish the pros/cons of using FaST. The comparison against CoVLM and CogCOM appears more significant (Fig 4) but is missing a comparison to the LLaVA-1.5 baseline w/ the stronger encoder.

– [Analysis] The results presented in Figure 5 are very interesting, but are missing baseline performance of the base LLaVA-1.5 model: what is its accuracy on the subset of system 2 / system 1 questions (as judged by FaST) in GQA/MME? How much does “slow” thinking actually improve performance on such questions (assuming that the switch adapter is accurate – it would also be good to validate this eg. by collecting question complexity labels)?

– [Analysis] It would be valuable to include an evaluation of the correctness/quality of the “chain-of-evidence” generated by the approach, perhaps via a human study and/or model interpretability methods. Does the chain-of-evidence appear faithful or is it mostly a post-hoc rationale? How often does a “correct” chain of evidence lead to the right conclusion, and vice versa? Is the chain-of-evidence used for the model “optimal”, or does it include unnecessary steps or logical leaps? This will help build confidence around the improved transparency claims made in the paper.

**Questions:**

Please see weaknesses above. I believe the paper in its current form already presents promising ideas+results but can be made still stronger.

– It would be good include a discussion / conceptual comparison to recent “large reasoning models” that propose scaling test-time compute eg. the OpenAI o1 model. While not (yet) multimodal, such methods are speculated to be trained w/ reinforcement learning and to produce “internal” (i.e. not necessarily interpretable) chain of thought – what are some of the tradeoffs of this type of approach against FaST?

– [Minor] As I understand, FaST does not provide increased transparency when it opts for “fast” thinking – it would be good to clearly demarcate this.

– [Minor] L538 typo: extra "A"

---

> ### Author Response · Authors · 2024-11-20
> **Response to Reviewer M2ti (1/3): Response to the Weaknesses Part I**
>
> We appreciate that the reviewer finds our approach novel and clear. Here are our responses:
>
> >1. [Complexity] One concern I have is the complexity of the approach, especially considering the large number of moving parts. Do the same set of method hyperparameters generalize across settings, or was a large amount of tuning required to achieve the results reported in the paper?
>
> Thank you for raising this concern on the complexity of our method. For the VQA and multimodal benchmarks, we utilize the same dataset for training, ensuring consistency and fairness in evaluation. For segmentation tasks, we carefully exclude datasets containing referring segmentation and reasoning segmentation validation and testing samples during training. This step is crucial to prevent potential data leakage, ensuring that the reported performance reflects the model's true generalization capability rather than memorization of specific datasets. So in conclusion, our methods can generalize across different settings.
>
> >2. [Clarity] I found some of the method details to be lacking eg. L259-260 mentions “a fast sampler based on cross-attention” but I could not find any further details for this component
>
> Thank you for pointing it out. Our approach employs a Perceiver Resampler, inspired by Flamingo [1], to efficiently process visual features. It utilizes learned latent queries that cross-attend to flattened visual features enhanced with temporal position encodings. This method significantly reduces the number of tokens generated by the original image encoder, optimizing computational efficiency. Additional details have been included in the Appendix A for further clarification.
>
> [1] Alayrac et al., 2022. Flamingo: a Visual Language Model for Few-Shot Learning.
>
> >3.1 [Performance] While FaST does seem to consistently outperform the LLaVA-1.5 base approach that it builds on top of, it is not exactly an apples-to-apples comparison considering the additional models / datasets that are used (including large ones such as SAM).
>
> Thank you for pointing out the comparison challenges. While we acknowledge that in the VQA setting, the comparison may not be entirely apples-to-apples due to differences in datasets and pretraining models, our segmentation experiments provide a more direct evaluation. Specifically, we have compared FaST against LLaVA equipped with a Seg Adapter, and our method consistently outperforms LLaVA across multiple referring and reasoning segmentation benchmarks (as shown in Table 2 in our paper). Below are the results cropped from Table 2 on different segmentation task in CIoU Metric:
>
> | Method | refCOCO | refCOCO+ | refCOCOg | ReasonSeg |
> | :---- | :---- | :---- | :---- | :---- |
> | LLaVA w Seg | 70.8 | 57.5 | 64.0 | 43.0 |
> | FaST | **73.3** | **64.4** | **67.0** | **47.6** |
>
> These results demonstrate the effectiveness of our framework in handling segmentation tasks, particularly in reasoning segmentation where FaST achieves a notable improvement.
>
> > 3.2 [Performance] Meanwhile, its gains over methods reasoning-focused approaches like Visual CoT and Chain-of-Spot are modest (and sometimes lags considerably eg. behind VCoT on VQAT) – a deeper analysis of / comparison against these methods (eg. average runtime etc.) would be helpful to establish the pros/cons of using FaST.
>
> Regarding the performance of Visual CoT, its higher accuracy is partly due to its reliance on a larger and more curated dataset, which provides it with an advantage VQA tasks. However, when considering runtime efficiency, FaST demonstrates significant improvements. For example, the average runtime of FaST on VQA dataset is 1248ms, which is substantially faster than both Chain of Spot (1513ms) and Visual CoT (1524ms).
>
> | Methods | Runtime |
> | :---- | :---- |
> | Chain of Spot | 1513ms |
> | Visual CoT | 1524ms |
> | FaST | 1248ms |
>
> Additionally, VQA tasks generally involve lower difficulty and complexity compared to reasoning segmentation, making FaST particularly well-suited for real-world, daily-use applications where efficiency and scalability are critical. As foundation models continue to improve, we anticipate that System 1 reasoning will handle an increasing number of queries effectively, further reducing reliance on System 2 processing. This scalability highlights FaST’s potential for greater adaptability and efficiency in diverse settings as foundational models advance.

---

> ### Author Response · Authors · 2024-11-20
> **Response to Reviewer M2ti (2/3): Response to the Weaknesses Part II**
>
> **Cont.**
> > 3.3 [Performance] Meanwhile, its gains over methods reasoning-focused approaches like Visual CoT and Chain-of-Spot are modest (and sometimes lags considerably eg. behind VCoT on VQAT) – a deeper analysis of / comparison against these methods (eg. average runtime etc.) would be helpful to establish the pros/cons of using FaST.
>
> Thank you for your observation regarding Figure 4. We have updated the figure to include the performance of LLaVA with the stronger encoder for comparison. As shown in the revised figure, LLaVA achieves the lowest performance among CoVLM, CogCoM, and FaST, further demonstrating the effectiveness of our approach. FaST outperforms LLaVA by a significant margin while maintaining competitive results with other state-of-the-art methods, validating the advantages of our framework in both reasoning and segmentation tasks. This additional comparison strengthens the evidence supporting FaST’s robustness and scalability.
>
> > 4 [Analysis] The results presented in Figure 5 are very interesting, but are missing baseline performance of the base LLaVA-1.5 model: what is its accuracy on the subset of system 2 / system 1 questions (as judged by FaST) in GQA/MME? How much does “slow” thinking actually improve performance on such questions (assuming that the switch adapter is accurate – it would also be good to validate this eg. by collecting question complexity labels)?
>
> Thank you for the thoughtful feedback on baseline performance analysis. We have added a breakdown of System 1 and System 2 performance on VQA datasets, including results for the base LLaVA-1.5 model. Our updated analysis shows that FaST achieves comparable performance to LLaVA-1.5 on System 1 questions, demonstrating that the incorporation of System 2 reasoning does not degrade performance on straightforward tasks. More importantly, our method exhibits significant improvements over LLaVA-1.5 on System 2 questions, highlighting the effectiveness of “slow thinking” in addressing complex reasoning problems. This analysis underscores the value of our approach in enhancing performance on tasks that require deeper contextual and logical reasoning.
>
> | $VQA^{v2}$  | LLM  | System 1 | System 2 |
> | :---- | :---- | :---- | :---- |
> | BLIP-2 | 13B | 67.3 | 53.1 |
> | LLaVA-v1.5 | 7B | 81.2 | 68.0 |
> | Chain of Spot | 7B | 82.1 | 74.5 |
> | FaST | 7B | 81.1 | **75.5** |
>
> | GQA  | LLM  | System 1 | System 2 |
> | :---- | :---- | :---- | :---- |
> | BLIP-2 | 13B | 37.8 | 22.4 |
> | LLaVA-v1.5 | 7B | 70.3 | 47.0 |
> | Chain of Spot | 7B | 70.9 | 50.7 |
> | FaST | 7B | 70.2 | **52.3** |
>
> | $VQA^{T}$  | LLM  | System 1 | System 2 |
> | :---- | :---- | :---- | :---- |
> | BLIP-2 | 13B | 44.3 | 39.7 |
> | LLaVA-v1.5 | 7B | 61.1 | 53.7 |
> | Chain of Spot | 7B | 62.1 | 59.0 |
> | FaST | 7B | 61.2 | **60.2** |
>
> | $SQA^{I}$   | LLM  | System 1 | System 2 |
> | :---- | :---- | :---- | :---- |
> | BLIP-2 | 13B | 63.4 | 59.2 |
> | LLaVA-v1.5 | 7B | 68.4 | 65.7 |
> | Chain of Spot | 7B | 68.6 | 67.8 |
> | FaST | 7B | 68.2 | **70.2** |
>
> > 5 [Analysis] It would be valuable to include an evaluation of the correctness/quality of the “chain-of-evidence” generated by the approach, perhaps via a human study and/or model interpretability methods. Does the chain-of-evidence appear faithful or is it mostly a post-hoc rationale? How often does a “correct” chain of evidence lead to the right conclusion, and vice versa? Is the chain-of-evidence used for the model “optimal”, or does it include unnecessary steps or logical leaps? This will help build confidence around the improved transparency claims made in the paper.
>
> Thank you for the suggestion regarding the evaluation of the correctness and quality of the “chain-of-evidence” generated by our approach. To address this, we have included an analysis of failure cases in Appendix Figure 9, which illustrates four key scenarios where the model’s reasoning process breaks down. These include:
> 1. The model failing to trigger the \textit{System 2} thinking mode, which is critical for complex reasoning tasks.
> 2. The inability to construct adequate contextual clues, leading to incomplete or irrelevant guidance during the reasoning process.
> 3. Failure to generate appropriate proposals for potential solutions, undermining the progression of the chain-of-evidence.
> 4. Errors in providing accurate pixel masks, which are essential for grounding the reasoning in the visual domain.
>
> These cases highlight areas where the “chain-of-evidence” deviates from being faithful or optimal, offering insights into when and why the reasoning may include unnecessary steps or logical leaps. This analysis helps clarify the limitations and informs future improvements, supporting the transparency claims made in the paper.

---

> ### Author Response · Authors · 2024-11-20
> **Response to Reviewer M2ti (3/3): Response to the Questions Part**
>
> **Questions part:**
>
> > 1 It would be good include a discussion / conceptual comparison to recent “large reasoning models” that propose scaling test-time compute eg. the OpenAI o1 model. While not (yet) multimodal, such methods are speculated to be trained w/ reinforcement learning and to produce “internal” (i.e. not necessarily interpretable) chain of thought – what are some of the tradeoffs of this type of approach against FaST?
>
> Thank you for your thoughtful question. We acknowledge that the current version of FaST does not scale test-time compute in the manner of recent “large reasoning models,” such as the OpenAI o1 model. While these models leverage reinforcement learning and internal chains of thought to achieve scalability, they often require significant computational resources, making them less efficient.
>
> In contrast, FaST is designed to prioritize multimodal reasoning with a clear focus on transparency and adaptability. The use of interpretable reasoning modes (System 1 and System 2) ensures that FaST provides insights into its decision-making processes, which is critical for applications requiring explainability. Additionally, FaST’s modular design allows it to balance computational efficiency and accuracy dynamically, making it suitable for diverse and resource-constrained environments. These strengths highlight FaST’s unique contributions and its complementary potential to scalable reasoning approaches.
>
> We recognize the exciting opportunities presented by scalable reasoning models and are actively exploring ways to integrate such methodologies into FaST’s framework in future iterations, aiming to combine the best of both worlds—scalability and transparency. We have added this discussion in the Appendix Section F.
>
> > [Minor] As I understand, FaST does not provide increased transparency when it opts for “fast” thinking – it would be gcood to clearly demarcate this.
>
> Thank you for pointing this out. We acknowledge that FaST does not inherently provide increased transparency when operating in the “fast” thinking mode. To address this limitation, we have clearly demarcated the transparency differences between “fast” and “slow” modes in Introduction of the revised manuscript.
>
> > [Minor] L538 typo: extra "A"
> Thank you for pointing this out. The typographical error at Line 538, specifically the extra ‘A,’ has been corrected in the revised manuscript.
>
> We hope our response addresses your concerns. Please let us know if there are any additional questions, and we are happy to discuss further.

---

> > ### Comment · Reviewer_M2ti · 2024-11-25
> > **Thanks for the response**
> >
> > The author response has addressed most of my concerns, and the additional fine-grained performance analysis is compelling. I will retain my score recommending acceptance.

---

> > > ### Author Response · Authors · 2024-11-25
> > >
> > > Thank you for your prompt response. We are genuinely grateful for your thoughtful feedback. We are really appreciative of the review, as they clearly strengthen the completeness, and further illuminate the future direction of our work.
> > >
> > > Best,
> > >
> > > Authors

---

### Author Response · Authors · 2024-11-21
**Response to all Reviewers**

To all reviewers:

Thank you for your thorough review and insightful comments. We have revised our paper according to the suggestions. The major changes are summarized as follows:
1. [**Performance Evaluation**] As suggested by Reviewer M2ti, we performed additional experiments to compare System 1 and System 2 performance in FaST, detailed in Appendix Sec. B. The results demonstrate that FaST achieves significant improvements on System 2 tasks while maintaining competitive performance on System 1 tasks.
2. [**Dataset Construction and Analysis**] In response to Reviewer GSxm and uXT1, we added a detailed description of the dataset construction process in Appendix Sec. A, including methodologies for dataset synthesis and augmentation. Additionally, we confirmed plans to release all datasets upon publication.
3. [**Failure Case Analysis**] Addressing Reviewer GSxm and KNeJ, we included a comprehensive analysis of failure cases in Appendix Figure 9, identifying key scenarios where the reasoning pipeline underperforms. This provides insight into limitations and areas for improvement.
4. [**Clarity and Transparency**]
Per Reviewer M2ti’s feedback, we clarified details regarding the fast sampler mechanism, which is elaborated in Appendix Sec. A. We also revised descriptions of System 2 accuracy to avoid ambiguities and ensured consistency with Figure 5. Transparency differences between System 1 and System 2 were further emphasized in the revised Introduction.
5. [**Switch Adapter Performance**]
As suggested by Reviewer uXT1, we evaluated the efficiency and robustness of the switch adapter. Results in Appendix Sec. B show minimal computational overhead, validating its lightweight and effective design.
6. [**Extended Experiments**] Following Reviewer KNeJ’s suggestion, we conducted experiments using the Bongard problem to evaluate FaST’s generalization capabilities for abstract reasoning tasks. These results are included in Appendix Sec. C and demonstrate incremental improvements over baseline models.

All modifications have been marked in blue in our revised submission.

Sincerely yours,

Authors.

---

### Meta-Review · Area_Chair_acJv · 2024-12-21

**Metareview:**

This work presented a new reasoning mechanism which can adaptively switch between "system 1" and "system 2" for multimodal LLMs. The authors proposed a top-down reasoning pipeline for visual agent which starts from the global view of the visual input and gradually drill down to the detailed visual observatrions to derive the final outputs, either for visual question answering or reasoning segmentation. To make the visual agent adaptively, the authors proposed a switch module and train it with carefully curated data to enable automatic. The experimental results demonstrate that the proposed system-1/2 visual agent achieve the best of both world for both simple tasks that do not require long-run reasoning and complicated tasks that necessate the reasoning process. Some further visualizations can clearly demonstrate the reasoning steps which enhances the interpretability substantially.

The main contribution of this work is the proposed fast and slow thinking mechanism for the multimodal LLMs. The paper is well-written and movtivated. The method is also appeadling and systematic per the comments by reviewers. Overall, the ACs think this work is a great way of enhancing the explicit reasoning capability for multimodal LLMs, which mirrors the current tremendous efforts to improve the reasoning capability in language domains for coding and mathematical tasks.

After the engaging discussion period, most of the concerns raised by the reviewers were addressed by the authors. From the ACs' perspective, the point made by reviewer GSxm that "The proposed framework primarily focuses on VQA and segmentation tasks. To better mirror System 2 recognition, the reviewer suggests evaluating results from more complex reasoning tasks." makes much sense as image QA and segmentation tasks are mostly perception tasks without reliance on complicated reasoning as also pointed out by the authors and other reviewers. To really make the vision model as an agent as claimed by "Visual Agents", it is supposed to come with more  capable system-1/2 thinking for more agentic tasks.

Overall, three reviewers gave positive ratings on this submission, and one reviewer gave 5 but with most of the concerns addressed according to the reply. The ACs also agree that this is a nice work on a systematical way of enabling system-1/2 thinking for multimodal LLMs, despite it should not count as a visual agent per se. As a result, the ACs recommend a clear acceptance of this work.

**Additional Comments On Reviewer Discussion:**

Most of the concerns were addressed during the discussion period. Some minor points are remained but it does not affec the final decision.

---

### Decision · Program_Chairs · 2025-01-22

Accept (Poster)